# The phosphatase PTEN links platelets with immune regulatory functions of mouse T follicular helper cells

Xue Chen [1✉], Yanyan Xu[2], Qidi Chen [2], Heng Zhang [3], Yu Zeng[3], Yan Geng[2], Lei Shen [3], Fubin Li [3], Lei Chen [3], Guo-Qiang Chen [2], Chuanxin Huang [3✉] & Junling Liu [2✉]

Beyond a function in hemostasis and thrombosis, platelets can regulate innate and adaptive immune responses. Hyperactive platelets are frequently associated with multiple human autoimmune diseases, yet their pathogenic functions in these diseases have not been fully established. Emerging studies show an essential function of the phosphatase and tensin homolog (PTEN) in maintenance of immune homeostasis. Here, we show that mice with platelet-specific deletion of *Pten*, develop age-related lymphoproliferative diseases and humoral autoimmunity not seen in wildtype animals. Platelet-specific *Pten*-deficient mice have aberrant T cell activation, excessive T follicular helper (Tfh) cell responses and accumulation of platelet aggregates in lymph nodes. Transferred *Pten*-deficient platelets are able to infiltrate into the peripheral lymphoid tissues and form more aggregates. Moreover, *Pten*-deficient platelets are hyperactive and overproduce multiple Tfh-promoting cytokines via activation of the PDK1/mTORC2-AKT-SNAP23 pathway. *Pten*-deficient platelets show enhanced interaction with CD4$^+$ T cells and promote conversion of CD4$^+$ T cells into Tfh cells. Our results implicate PTEN in platelet-mediated immune homeostasis, and provide evidence that hyperactive platelets function as an important mediator in autoimmune diseases using mouse models.

[1] School of Life Sciences, Shanghai University, 333 Nanchen Road, Shanghai 200444, China. [2] Department of Biochemistry and Molecular Cell Biology, Key Laboratory of Cell Differentiation and Apoptosis of Chinese Ministry of Education, Shanghai Jiao Tong University School of Medicine, 280 South Chongqing Road, Shanghai 200025, China. [3] Shanghai Institute of Immunology, Department of Immunology and Microbiology, Shanghai Jiao Tong University School of Medicine, 280 South Chongqing Road, Shanghai 200025, China. ✉email: chenxue-snow@163.com; huangcx@shsmu.edu.cn; liujl@shsmu.edu.cn

Anucleate blood platelets are well-known to participate in hemostasis and thrombosis, and their aberrant activation is implicated in multiple human cardiovascular diseases. Platelets can also affect multiple types of innate and adaptive immune cells to regulate immune and inflammatory responses[1–3]. Platelets promote activation and differentiation of CD4[+] helper T cells[4,5], and induce immunoglobulin class-switching and B-cell antibody responses[6,7]. Platelets and megakaryocytes have a function in processing and presenting both foreign and self-antigens via MHC class I molecules, and regulate the activation and function of CD8[+] cytotoxic T-lymphocytes[8,9]. Platelets are involved in surveying and collaborating with macrophages via the platelet-adhesion receptor GPIb, rapidly helping eradicate blood-borne bacterial infection in the liver[10]. Platelets are able to regulate the maturation and cytokine secretion of dendritic cells (DCs), establishing the communication between innate and adaptive immune responses[6]. It has been believed that platelets are the most numerous circulating cell type harboring immune regulatory functions.

Platelets can produce over 1100 proteins as identified by proteomics[11]. Activated platelets release the contents of their a-granule and secrete a variety of cytokines, chemokines, growth factors, and inflammatory mediators, including P-selectin, TGF-β, IL-6, IL-21, IFN-γ, and non-neuronal serotonin[12–15]. The inflammatory cytokines IL-6 and IL-21 are critical for T follicular helper (Tfh) cell differentiation and proliferation[16–18], and IFN-γ promotes the generation and effector function of Th1 cells[19]. Moreover, platelets from the patients with systemic lupus erythematosus (SLE) displayed increased IFN pathway signatures and enhanced activation[15]. And also, platelets express and secrete a plethora of critical immune molecules such as CD40, CD40 ligand (CD40L), and MHC class I molecules, which allows them to directly influence innate and adaptive immune responses during the immune disease processes[9]. Of note, activated platelets are the major source of soluble CD40 ligand (sCD40L) in the circulation[20]. CD40 is expressed on multiple immune cells including CD8[+] T cells, CD4[+] T cells, B cells, and DCs[21–23]. The CD40L-CD40 interaction has been recognized to be of critical importance in adaptive immune response and inflammation in the infectious diseases like atherosclerosis[23,24]. Moreover, CD40-sCD40L ligation activates platelets and subsequently enhances platelet-leukocyte adhesion, which is important in the recruitment of leukocytes to sites inflammation[23].

In addition, platelets can physically interact with various immune cells by a variety of factors[25,26]. Most importantly, the factors involved in direct interactions include platelet P-selectin, CD40/CD40L and GPIb, as well as PSGL-1, CD40L/CD40 and Mac-1 (integrin αMβ2, CD11b/CD18) on other immune cells[10,27,28]. The direct interaction between platelets and immune cells is also important for their activation and functions[29]. Platelets can bind to CD40L-positive T cells, triggering granule release and recruiting T cells through the secreted RANTES[29]. Platelets form aggregates with monocytes via surface P-selectin, consequently inducing monocyte tissue factor expression in patients with severe COVID-19[27]. Platelets form aggregates with CD4 T cells via P-selectin and CD166, regulating pathogenic CD4 T-cell functions during autoimmune neuroinflammation[12]. Platelets form aggregates with Treg cells via sCD40L and P-selectin under infection conditions[28,30]. The aggregations alter the transcription profiles of Treg cells, promoting them to enter the lung to resolve pulmonary inflammation[28,30].

Hyperactive platelets are frequently observed in various human autoimmune diseases such as multiple sclerosis[31], rheumatoid arthritis, and systemic lupus erythematosus[32]. Compared with healthy individuals, the activation of platelets is increased in patients with SLE. These patients have increased levels of thromboxane and βthromboglobulin, an indicator of sustained platelet activation. In SLE patients, the increased levels of platelet surface P-selectin[33,34] and platelet-derived soluble CD40L are correlated with the SLE disease activity index (SLEDAI) score[35,36]. In addition, platelets from SLE patients are also more readily aggregated ex vivo than those from healthy donors, suggesting that in SLE, platelets are primed or hyperactive in vivo[32,34]. However, the cellular and molecular mechanisms of hyperactive platelets in regulating autoimmune diseases, such as SLE, have not been demonstrated.

Phosphatase and tensin homolog (PTEN) acts as a tumor suppressor frequently mutated in diverse cancers[37]. It is well established that PTEN regulates many cellular processes including survival, differentiation, energy metabolism, cellular architecture, and mobility[37]. Recently, it is increasingly clear that PTEN also has important functions in maintenance of immune homeostasis[37], although the underlying mechanisms remain to be further explored. Our previous study showed that platelets require PTEN to restrain collagen-induced activation and aggregation by inhibiting lipid kinase PI(3)K-AKT signaling, and extend the bleeding time[38]. Therefore, PTEN-deficient platelets are hyperactive and platelet-specific Pten-deficient (Pten[fl/fl]Pf4-Cre) mice are suitable to be used as an ideal model to explore the functions of hyperactive platelets in vivo.

In this work, we demonstrate that platelet-specific Pten-deficient mice develop lymphoproliferative and autoimmune diseases mainly by inducing excessive Tfh cell responses compared with wild-type controls. Mechanistically, Pten-deficient platelets oversecrete various Tfh-promoting cytokines through AKT-SNAP23 signaling compared to Pten-sufficient platelets. Pten-deficient platelets are able to infiltrate into lymphoid tissues and convert CD4[+] T cells into Tfh cells via physical contact. Our study establishes a potentially causative function of hyperactive platelets in autoimmunity. The linking of platelets to Tfh cells could provide a therapeutic approach for treating autoimmune diseases.

## Results

**Platelet-specific Pten-deficient mice develop age-related lymphoproliferative and autoimmune diseases.** To explore the function of hyperactive platelets in maintenance of immune homeostasis, we generated platelet-specific Pten-deficient (Pten[fl/fl]Pf4-Cre) mice. Pten[fl/fl]Pf4-Cre mice were undistinguishable from the wild-type littermates by 5 weeks of age. Beginning at 2 months, Pten[fl/fl]Pf4-Cre mice spontaneously developed lymphoid hyperplasia (Fig. 1a). Specifically, the peripheral lymph nodes (pLNs) in Pten[fl/fl]Pf4-Cre mice, especially those around the inguinal, axillary, and submandibular regions, were dramatically enlarged (Fig. 1a, b). Flow cytometric analysis showed the expansion of B220[+] B cells relative to CD4[+] T cells in lymph nodes from these mice (Fig. 1c). However, the absolute numbers of both B220[+] B cells and CD4[+] T cells were increased in these lymphoid tissues (Fig. 1d). Interestingly, the relative expansion of B cells was only observed in lymph nodes from Pten[fl/fl]Pf4-Cre mice, but not in bone marrow, spleen and peripheral blood (Supplementary Fig. 1a, c). Moreover, Pten loss in platelets had no effect on B-cell differentiation in bone marrow (Supplementary Fig. 1b, d).

Pten[fl/fl]Pf4-Cre mice had significantly higher amounts of serum IgG and IgM compared to age-matched wild-type (WT) controls (Fig. 1e). Moreover, these mice generated higher concentrations of antibodies to double-stranded DNA (anti-dsDNA) in their plasma (Fig. 1f), a hallmark of autoimmune disease in human. Autoimmunity is frequently associated with renal pathology. Indeed, the glomeruli in the kidney from aged Pten[fl/fl]Pf4-Cre

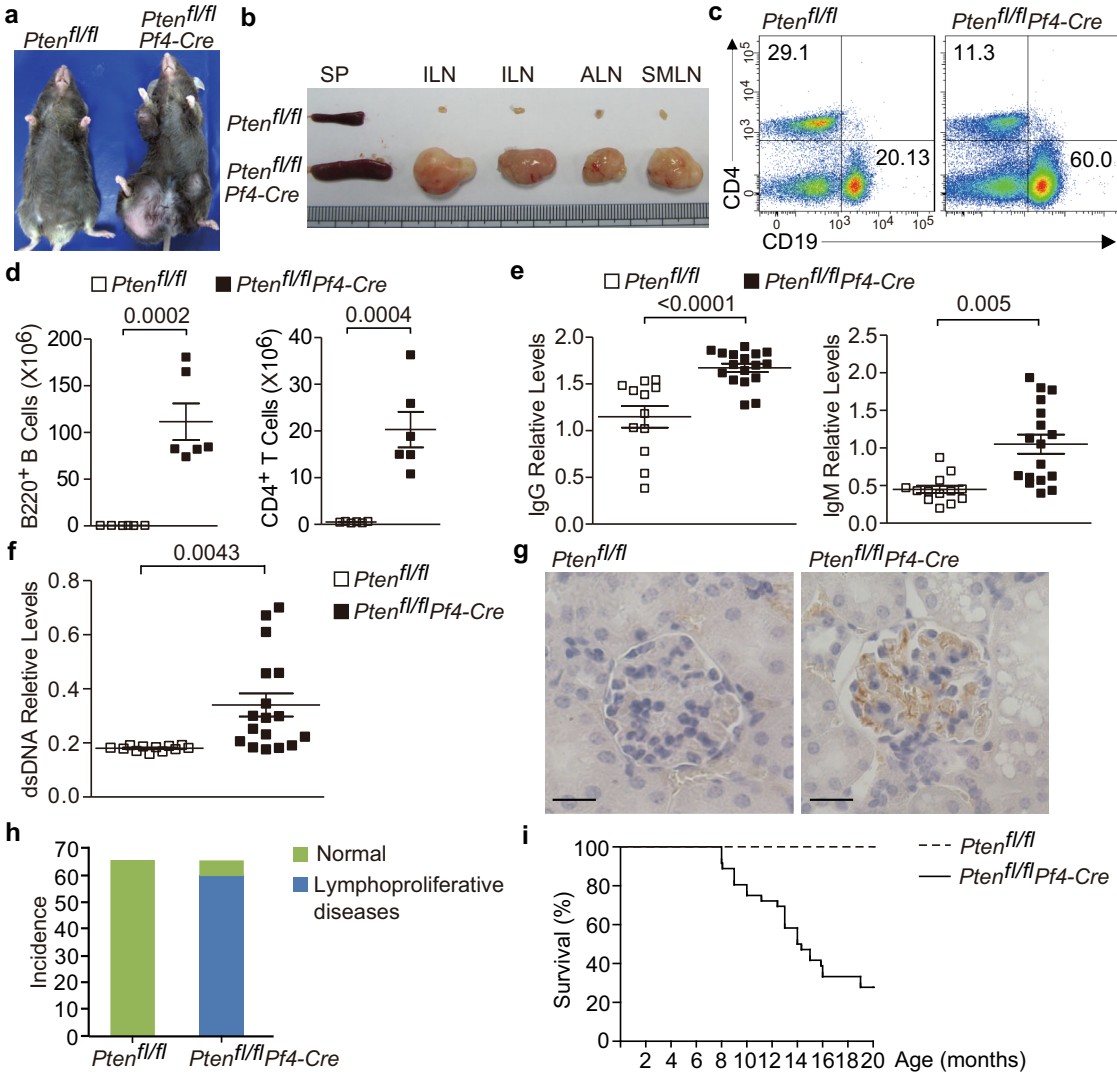

**Fig. 1 $Pten^{fl/fl}Pf4$-Cre mice develop age-related autoimmune and lymphoproliferative diseases.** Around 8-month-old $Pten^{fl/fl}$ and $Pten^{fl/fl}Pf4$-Cre mice were subjected for analysis (**a–f**). **a** The appearance of aged $Pten^{fl/fl}Pf4$-Cre mice with lymphoid hyperplasia. **b** Spleens and pLNs from $Pten^{fl/fl}$ and $Pten^{fl/fl}Pf4$-Cre mice with lymphoproliferative diseases. Spleen, SP; inguinal lymph node, ILN; axillary lymph node, ALN; and submaxillary lymph node, SMLN. Representative flow cytometry plots (**c**) and absolute numbers (**d**) of B220$^+$ B and CD4$^+$ T cells in pLNs from $Pten^{fl/fl}$ and $Pten^{fl/fl}Pf4$-Cre mice, respectively ($n = 6$). **e** Titers of immunoglobulins in plasma of $Pten^{fl/fl}$ (IgG, $n = 12$; IgM, $n = 13$) and $Pten^{fl/fl}Pf4$-Cre mice (IgG, $n = 17$; IgM, $n = 17$). **f** Titers of anti-dsDNA antibodies in plasma of $Pten^{fl/fl}$ ($n = 12$) and $Pten^{fl/fl}Pf4$-Cre mice ($n = 17$). All sex-matched mice were analyzed at the age of 8 months (**a–f**), and each symbol represents one mouse. **g** Immunochemistry of IgG deposit in renal glomeruli (Scale bars, 10 μm). Frequencies of $Pten^{fl/fl}Pf4$-Cre mice which showed lymphoid hyperplasia (**h**), and the survival curve of $Pten^{fl/fl}$ and $Pten^{fl/fl}Pf4$-Cre mice (**i**). $Pten^{fl/fl}$ and $Pten^{fl/fl}Pf4$-Cre mice ($n = 66$) were monitored for the appearance of lymphoid hyperplasia and survival within 2 years. Each symbol represents one mouse, and data shown in (**d–f**) are presented as mean ± s.e.m. (two-tailed $t$ test). Results are representative of at least three independent experiments. Source data are provided as a Source Data file.

mice displayed prominent IgG deposits (Fig. 1g). Up to two years, about 58 out of 66 $Pten^{fl/fl}Pf4$-Cre mice (87.9%) eventually developed severe lymphadenopathy (Fig. 1h and Supplementary Fig. 1e), which resulted in diminished survival (Fig. 1i). Hence, platelets require PTEN to limit lymphoproliferative disease and autoimmunity.

**Increased numbers of Tfh and GC B cells in $Pten^{fl/fl}Pf4$-Cre mice compared to controls.** We further explored whether $Pten$-deficient platelets influenced immune homeostasis. We observed that $Pten^{fl/fl}Pf4$-Cre mice had greater numbers of the CD62L$^{lo}$CD44$^{hi}$ population with an activated or memory phenotype in the CD4$^+$ and CD8$^+$ compartments of pLNs (Fig. 2a, b). The proportions of Th1, Th2, and Th17 cells, but not Foxp3$^+$

regulatory T cells, were significantly increased in the pLNs from these mice (Fig. 2c, Supplementary Fig. 2). In addition to these conventional effector T cells, we observed a marked expansion of CXCR5$^+$BCL6$^+$ or CXCR5$^+$PD1$^+$ Tfh cells in the pLNs of $Pten^{fl/fl}Pf4$-Cre mice (Fig. 2d). Accordingly, GL7$^+$Fas$^+$ GC B cells had also expanded in these lymphoid tissues (Fig. 2e). Immunochemistry showed an increase in the number and size of germinal centers, which were indicated by peanut agglutinin (PNA)-positive clusters, in the pLNs of $Pten^{fl/fl}Pf4$-Cre mice (Fig. 2f). Consistent with these findings, we also noted a significant increase of CD138$^+$ plasma cells in the pLNs of $Pten^{fl/fl}Pf4$-Cre mice as compared with littermate controls (Fig. 2g). Aberrant T-cell activation and differentiation as well as excessive Tfh cell responses were also observed in the spleens of $Pten^{fl/fl}Pf4$-Cre mice

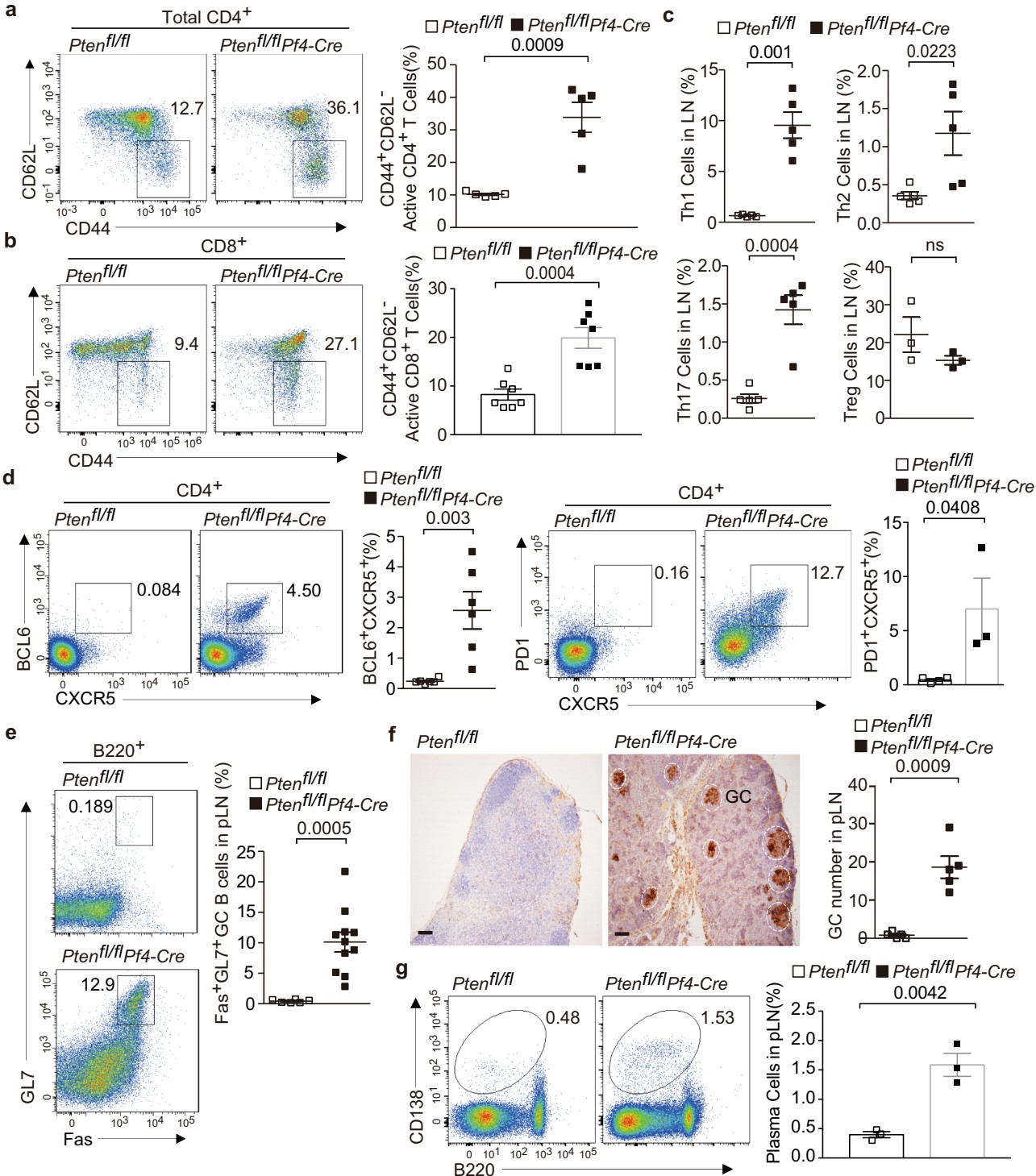

**Fig. 2 Altered T-cell differentiation and excessive Tfh cell responses in the pLNs of *Pten*^fl/fl*Pf4-Cre* mice.** The pLNs of 6-month-old mice were subjected for analysis (**a–g**). Flow cytometry plots and quantifications of CD62L⁻ CD44⁺ activated CD4⁺ T cells (**a**, *n* = 5) or CD8⁺ T cells (**b**, *n* = 7). **c** Flow cytometry analysis and quantifications of proportions of IFN-γ + (*n* = 5), IL-4⁺ (*n* = 5), IL-17⁺ (*n* = 5) and Foxp3⁺ cells (*n* = 3) among CD4⁺ T cells. **d** Flow cytometry analysis and quantifications of proportions of CXCR5⁺BCL6⁺ (*n* = 6) or CXCR5⁺PD-1⁺ Tfh cells (*Pten*^fl/fl *n* = 4; *Pten*^fl/fl*Pf4-Cre n* = 3) among live CD4⁺ T cells. **e** Flow cytometry analysis and quantifications of proportions of Fas⁺GL7⁺ GC B cells among live B220⁺ cells (*Pten*^fl/fl *n* = 6; *Pten*^fl/fl*Pf4-Cre n* = 11). **f** Representative PNA staining of lymphoid sections (scale bars, 100 μm) (*n* = 5). Germinal centers (GC) are identified as PNA-positive clusters and circled by while line. The number of GCs per section of pLNs (right) was shown at the right. **g** Flow cytometry plots and quantifications of CD138⁺ plasma cells (*n* = 3). Each symbol represents one mouse (**a–g**), the experiments were independently repeated at least three times and data are presented as mean ± s.e.m (**a–g**). ns, not significance and (two-tailed *t* test). Source data are provided as a Source Data file.

(Supplementary Fig. 3). Platelets have been reported to modulate the phenotypes of DCs in some cases[39], and DCs are critical for initiation of Tfh cell differentiation. We observed that *Pten*fl/fl*Pf4-Cre* mice at the age of 3–5 months displayed excess Tfh cells, but normal numbers, maturation and activation of DCs in the spleens (Supplementary Fig. 4). These results suggest that excess Tfh cells are not secondary effect of deregulated DCs in the context of *Pten*-deficient platelets. Together, these results demonstrate that platelets require PTEN to restrain excess Tfh cell responses and further maintain immune homeostasis.

**Aberrant T-cell activation and Tfh cell responses are cell-extrinsic.** PF4-Cre-mediated recombination can also occur in a small population of non-megakaryocytes /platelets lineages under some conditions[40]. PTEN deficiency has reported to render the competition advantage of multiple immune cells in some conditions[41,42]. Using *Rosa26*mT/mG*Pf4*-Cre reporter mice in which Cre-mediated removal of STOP cassette activates GFP expression, we found that very low percentages of CD4+ T and CD19+ B lineages (less than 5%) were positive for PF4-Cre (GFP +) in the pLNs and spleens from wild-type *Rosa26*mT/mG*Pf4*-Cre mice as well as *Rosa26*mT/mG*Pten*fl/fl*Pf4-Cre* mice (Supplementary Fig. 5). Besides, PF4-Cre-mediated recombination didn't occur among Treg cells (Supplementary Fig. 5). A small population of Mac1+ monocytes and F4/80+ monocytes/macrophages were GFP-positive, and the frequencies of GFP+ population significantly increased in the absence of *Pten* (Supplementary Fig. 5). However, PTEN deficiency in myeloid and mast cells had been reported to restrain inflammation, autoimmunity and Tfh cell responses[43–46]. Moreover, *Pten*-deficient macrophages and DCs are shown to protect mice from autoimmune diseases like experimental autoimmune encephalomyelitis through abrogating pathogenic Th17 cells polarization in autoimmunity[43]. Therefore, nonspecific *Pten* loss in these cells appear to partially counteract the autoimmunity observed in *Pten*fl/fl*Pf4-Cre* mice.

We next sought to determine whether the small population of *Pten*-deficient T and B lymphocytes expanded and dominated activated T cells, Th1 and Tfh subsets as well as GC B-cell lineages in *Pten*fl/fl*Pf4-Cre* mice. The frequencies of GFP+ cells among activated T cells, Th1 and Tfh subsets as well as GC B cells were less than 5%, and the numbers have not significantly changed in *Rosa26*mT/mG*Pten*fl/fl*Pf4-Cre* mice compared to the *Rosa26*mT/mG*Pf4*-Cre littermates (Supplementary Fig. 6), indicating that contribution of nonspecific deletion of *Pten* in T/B subsets is negligible in this setting. Meanwhile, the expansion of activated, Th1 and Tfh subsets as well as GC B cells occurred in GFP- *Pten*-sufficient compartments, suggesting that this effect is cell-extrinsic (Supplementary Fig. 7).

***Pten*-deficient platelets promote CD4+ T cells to gain Tfh signatures.** We next sought to investigate whether CD4+ T cells are skewed toward Tfh cells in the presence of *Pten*-deficient platelets. We compared the transcriptional profiles of CD4+ T cells isolated from *Pten*fl/fl*Pf4-Cre* mice and *Pten*fl/fl littermates. CD4+ T cells from *Pten*- deficient mice pLNs contained a total of 602 genes with greater than twofold difference, including 357 upregulated and 245 downregulated genes. Gene ontology was next performed to analyze the differentially expressed genes at the twofold cut-offs (Fig. 3a). The upregulated genes in CD4+ T cells from *Pten*fl/fl*Pf4-Cre* mice implicated in multiple autoimmune diseases including systemic lupus erythematosus (Fig. 3a). We selected genes from published data sets (GEO accession codes GSE21379) that are upregulated in Tfh cells relative to their expression in non-Tfh cells, for gene-set-enrichment analysis (GSEA) with our data. This analysis showed that the Tfh gene

signatures were significantly overrepresented in CD4+ T cells from *Pten*fl/fl*Pf4-Cre* mice (Fig. 3b). CD4+ T cells from these mice contained greater amounts of transcripts of genes involved in Tfh differentiation, including *Pdcd1, Cxcr5, Irf4, Il21, Btla, Il4,* and *Bcl6* (Fig. 3c)[47,48].

**PTEN deficiency enhanced platelet activation and over-produced Tfh-associated cytokines.** *Pten* deletion had no effect on platelet counts in peripheral blood or megakaryocyte ratio in bone marrow (Fig. 4a). And also, α-granule counts and P-selectin expression were both normal in *Pten*-deficient platelets (Supplementary Fig. 8a, b). P-selectin resides at the α-granule membrane in resting platelets and could translocate to plasma membrane via α-granule secretory pathway upon activation. Thus, the abundance of cell membrane-located P-selectin reflects the extent of α-granule release[49]. In response to thrombin, *Pten*-deficient platelets displayed higher levels of P-selectin on their surface compared with the wild type (Fig. 4b), indicating that PTEN deficiency resulted in hypersecretion of α-granules. In consistency with our previous reports[38], *Pten*-deficient platelets are hyper-active, evidenced by the fact that they formed larger aggregates in response to thrombin, collagen, or ADP (Fig. 4c).

Upon activation, platelets can release more than 300 active substances such as cytokines, chemokines, and growth factors[50]. Among them, 46 cytokines were reported to have immunomodulatory activities. In consideration of the hypersecretion of α-granules caused by PTEN deficiency, we compared the levels of these cytokines in the supernatant between wild-type and *Pten*-deficient platelets in response to thrombin using cytokine array. Although the cytokine contents were normal in the resting *Pten*-lacked platelets (Supplementary Fig. 8c), the levels of several cytokines including IL-6, IL-17B/F, IL-21, and IFN-γ were significantly upregulated in the supernatant from activated *Pten*-deficient platelets of *Pten*fl/fl*Pf4-Cre* mice (Fig. 4d, Supplementary Fig. 8d, e). Accordingly, gene ontology and Kyoto Encyclopedia of Gene and Genomes (KEGG) pathway analysis showed the gene signatures activated by IL-6, IL-17 and IFN-γ were significantly overrepresented in CD4+ T cells from *Pten*fl/fl*Pf4-Cre* mice (Fig. 4e). These cytokines have been proven to regulate the differentiation and effector functions of T cells. In particular, IL-21 is secreted by Tfh cells during germinal center reactions and acts directly on germinal center B cells to promote their proliferation and differentiation[16]. Germinal center-located platelets might provide IL-21 to promote germinal center responses. IL-6 facilitates early Tfh cell differentiation and is required for germinal center formation[51]. Interestingly, B-cell-derived IL-6 initiates spontaneous autoimmune germinal center formation in Lupus-prone mice[18]. We speculate that IL-21 and IL-6 are important cytokines produced by *Pten*-deficient platelets to induce excessive germinal center responses.

***Pten*-deficient platelets overproduced Tfh-promoting cytokines via activation of the PDK1/mTORC2-AKT-SNAP23 pathway.** We further wanted to investigate the molecular mechanism underlying α-granule hypersecretion in *Pten*-deficient platelets. *Pten*-deficient platelets displayed more levels of PDK1-mediated AKT Thr308 phosphorylation, Sin1 phosphorylation (a component of mTORC2), mTORC2-mediated AKT Ser473 phosphorylation, and membrane-bound P-selectin than WT counterparts in response to thrombin (Fig. 4f, g), a potent platelet agonist. Interestingly, upon thrombin stimulation, *Pten*-deficient platelets also significantly enhanced the phosphorylation levels of SNAP23 at Ser95, which is critical for synaptic release and platelet secretion (Fig. 4f). Importantly, the inhibitors for PDK1, AKT, or mTORC2 could effectively reduce the activation of AKT, Sin1,

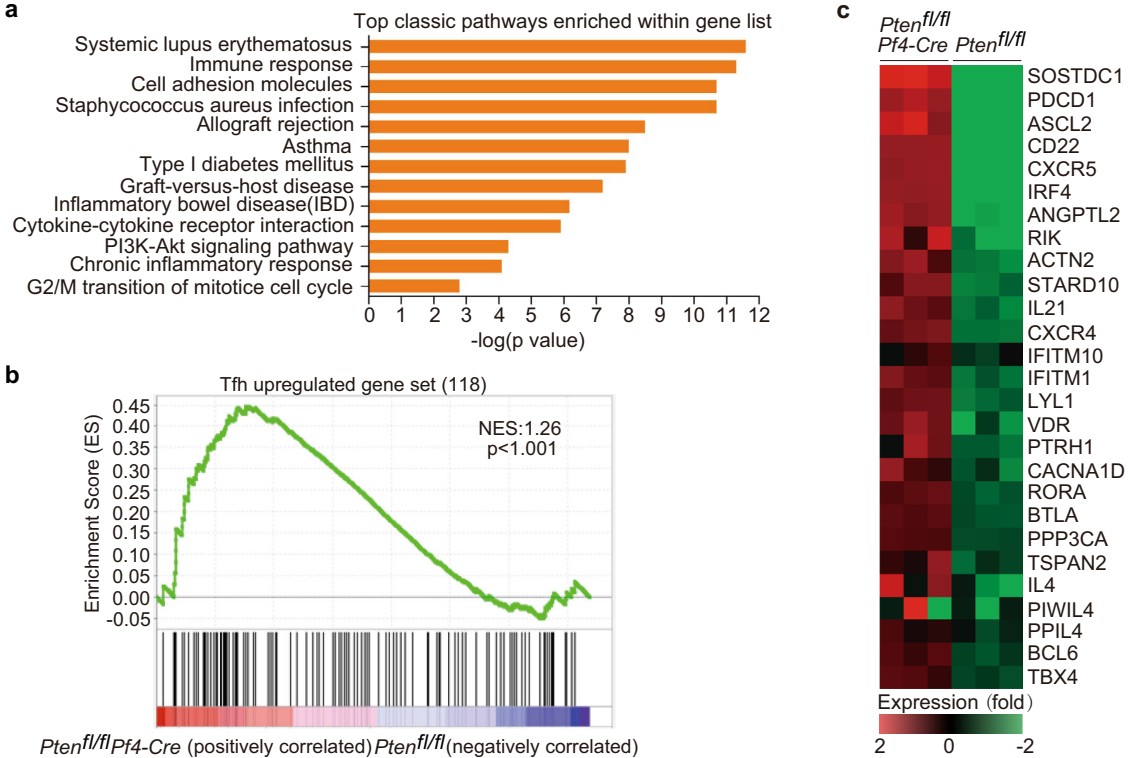

**Fig. 3 PTEN loss in platelets leads to the skewing of CD4⁺ T cells toward Tfh cells.** The pLNs of 6-month-old mice were subjected for RNA-Seq analysis (**a–c**). **a** Gene Ontology (GO) Categories enriched for differentially expressed genes between CD4⁺ cells of pLNs from $Pten^{fl/fl}$ and $Pten^{fl/fl}Pf4$-$Cre$ mice. **b** GSEA shows the Tfh-related genes as one of the most extensively upregulated pathways in $Pten^{fl/fl}Pf4$-Cre CD4⁺ T cells. One-sided permutation test was used to identify the significantly changed pathways. **c** Heat maps of differential gene expression, relative to the WT mean (≥2-fold), of Tfh-related genes. The scale below the heat maps indicates relative gene expression changes normalized by the s.d. (from −2.0 to +2.0 in log₂ units). Source data are provided as a Source Data file.

and SNAP23 by decrease their phosphorylation (Fig. 4f). Furthermore, the inhibitors for PDK1, AKT as well as mTORC2 could also efficiently reduce platelet granule secretion reflected by decreased membrane-bound P-selectin in *Pten*-deficient platelets treated by thrombin (Fig. 4g). Therefore, PTEN deficiency activates the PDK1/mTORC2(Sin1)-AKT axis, subsequently phosphorylating SNAP23 and enhancing α-granule secretion of platelets.

**_Pten_-deficient platelets enhance their ability of interacting with CD4⁺ T cells and promoting Tfh cell differentiation.** We next sought to ask whether *Pten*-deficient platelets regulate Tfh cell differentiation. To this end, fresh platelets were co-cultured with CD4⁺ T cells in vitro. Platelets were able to promote CD4⁺ T cells to express CXCR5, a hallmark of Tfh cells, and this effect was in dose-dependent manner (Fig. 5a). Interestingly, this capacity was significantly increased in the absence of *Pten* (Fig. 5a). Moreover, wild-type platelets pretreated with Bisperoxovanadium, a selective PTEN inhibitor, enhanced their ability in promoting CXCR5 expression in CD4⁺ T-cell under these conditions (Fig. 5b). Moreover, *Pten*-deficient platelets formed greater numbers of aggregates with CD4⁺ T cells than WT platelets under coculture conditions (Fig. 5c). Finally, flow cytometry showed that CD4⁺ T cells formed more aggregates with platelets in the pLNs of $Pten^{fl/fl}Pf4$-Cre mice (Fig. 5d). To further evaluate whether *Pten*-deficient platelets directly promote aberrant Tfh cell responses, we performed the platelet transfer experiments. Recipient mice generated more Tfh cells and GC B cells in the pLNs after transfer with *Pten*-deficient platelets than WT platelets

(Fig. 5e, f). In conclusion, these results indicate that *Pten*-deficient platelets increase the ability of interacting with CD4⁺ T cells and promoting Tfh cell responses.

**_Pten_-deficient platelets infiltrate into lymphoid tissue and form aggregates.** Platelets can extravasate and interact with fibroblastic reticular cells (FRCs) around high endothelial venule (HEV)[52]. The transmembrane O-glycoprotein Podoplanin, expressed on FRCs, acts as an activating ligand for platelet C-type lectin-like receptor 2 (CLEC-2). Activated platelets release sphingosine-1-phosphate (S1P) and further promote VE-cadherin expression on HEVs[52]. This axis-mediated platelet extravasation is critical for maintenance of HEV integrity[52]. Moreover, the released S1P, a bioactive metabolite of sphingosine, has a pivotal intrinsic function in platelet activation and can further activate platelets[53]. It is possible that excess extravasated platelets are activated and further infiltrate into the tissues. In recent, accumulating evidences have demonstrated that activated platelets infiltrate into tissues to regulate inflammatory responses[10,28,30]. We hypnotized that PTEN deficiency promoted platelets to infiltrate into the lymphoid tissues. We utilized the $Rosa26^{mTmG}Pf4$-$Cre$ reporter mice in which platelets are indicated by PF4-Cre driven GFP expression. We observed that $Rosa26^{mT/mG}Pten^{fl/fl}Pf4$-$Cre$ mice displayed an increase in the number and size of GFP-positive platelet aggregates in their lymph nodes, especially in the T-cell zones (Fig. 6a, b). Moreover, we observed that more *Pten*-deficient platelet aggregates interacted with CD4⁺ T cells (Fig. 6c). Interestingly, *Pten*-deficient platelet aggregates located to some germinal centers in the pLNs (Fig. 6d). To further investigate

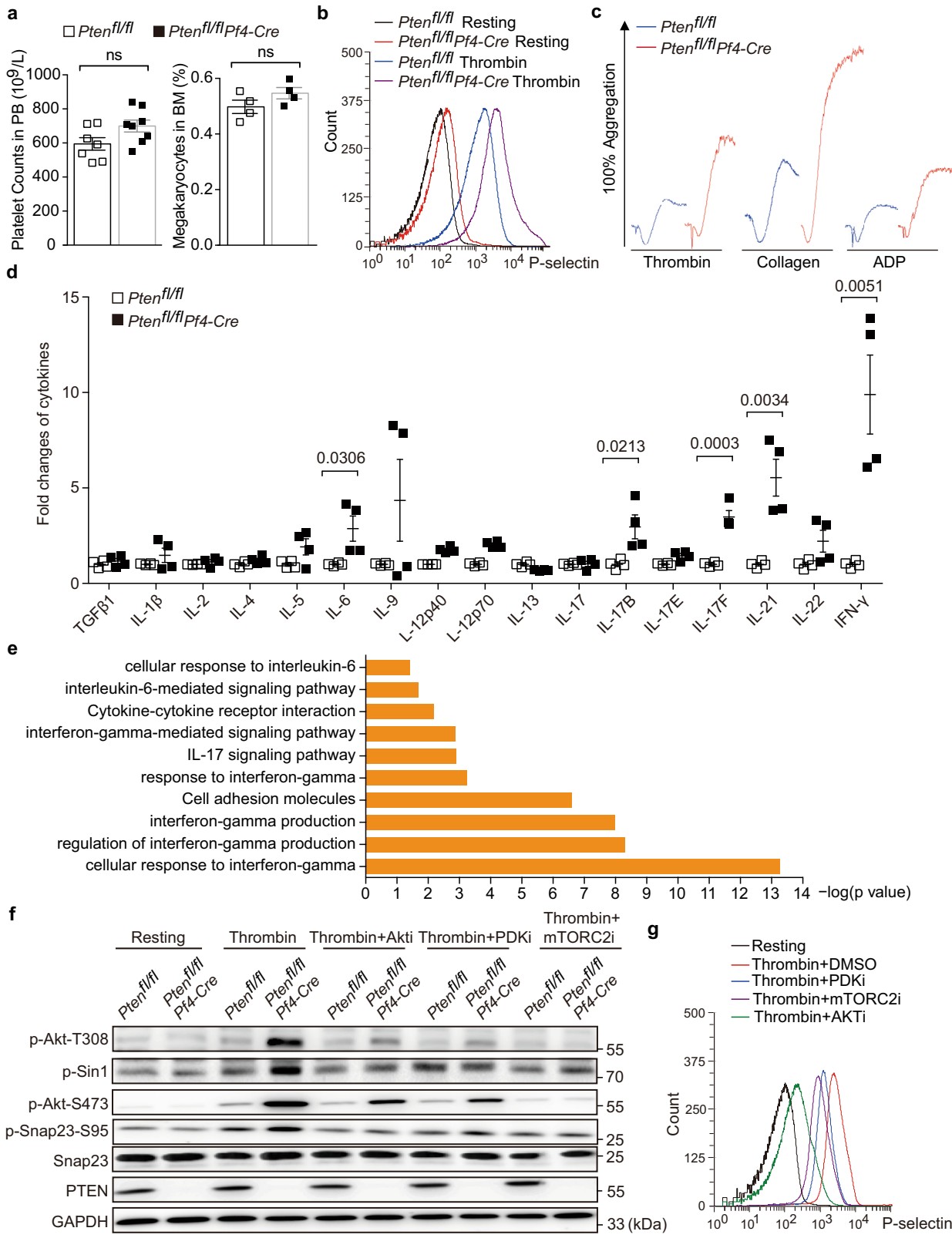

whether PTEN affects platelet infiltration and clustering in vivo, GFP-labeled WT (*Rosa26*^mT/mG*Pf4-Cre*) or *Pten*-deficient (*Rosa26*^mT/mG *Pten*^fl/fl*Pf4-Cre*) platelets were transferred into WT mice via tail vein injection (Fig. 6e). More *Pten*-deficient platelets accumulated and formed aggregates in the lymph nodes in an extravascular manner (Fig. 6f, g), indicating that *Pten*-deficient platelets gain the ability of entrance into lymphoid tissues.

## Discussion

Emerging studies have shown that hyperactive platelets are frequently associated with various human autoimmune diseases[32,34,54,55]. The importance of hyperactive platelets in the pathogenesis of autoimmune diseases has not been established. In this study, we deleted *Pten* in platelets to generate mice with hyperactive platelets. These mice developed lymphoid hyperplasia,

**Fig. 4 PTEN deficiency enhances platelet aggregation and cytokines secretion through activation of PDK1/mTORC2-AKT-SNAP23 pathway.** The pLNs of 6-month-old mice were subjected for analysis (**a–g**). **a** Platelet counts (left) and megakaryocyte (right) ratio in peripheral blood or bone marrow of *Pten*^fl/fl^ (platelet, $n = 7$; megakaryocyte, $n = 4$) and *Pten*^fl/fl^*Pf4-Cre* mice (platelet, $n = 8$; megakaryocyte, $n = 4$). **b** Representative flow cytometry analysis of the surface P-selectin on *Pten*^fl/fl^ and *Pten*^fl/fl^*Pf4-Cre* platelets stimulated with 0.1 U/ml Thrombin. **c** *Pten*^fl/fl^ and *Pten*^fl/fl^*Pf4-Cre* platelets aggregation in response to 0.1 U/ml Thrombin, 1 μg/ml Collagen and 10 μM ADP, respectively. **d** Relative levels of cytokines released from platelets of *Pten*^fl/fl^ and *Pten*^fl/fl^*Pf4-Cre* mice stimulated with 0.1 U/ml Thrombin ($n = 4$). Data are presented as fold changes relative to wild-type platelets. **e** The pathway signatures in response to IL-6, IL-17, and IFN-γ were significantly overrepresented in CD4^+^ T cells isolated from pLNs from *Pten*^fl/fl^*Pf4-Cre* mice. **f** Western blotting of indicated proteins in wild-type and *Pten*-deficient platelets in response to 0.1 U/ml Thrombin with the presence of DMSO or indicated inhibitors.
**g** Representative flow cytometry analysis of the surface P-selectin on *Pten*^fl/fl^ platelets stimulated with 0.1 U/ml Thrombin in the presence of DMSO or indicated inhibitors. Results are representative of three independent experiments. The data shown in (**a**, **d**) are presented as mean ± s.e.m. (two-tailed *t* test). Source data are provided as a Source Data file.

excessive Tfh cell responses, and spontaneous inflammatory diseases. Our work demonstrates a crucial function of PTEN for platelet to maintain immune homeostasis, and implies hyperactive platelets as an important mediator in human autoimmune diseases. Mutations and single nucleotide polymorphisms in PTEN have been reported in SLE patients by whole-exome sequencing[56,57]. It is likely that these genetic alterations may affect PTEN expression and function in platelets, subsequently causing platelet-mediated disruption of immune homeostasis. It is interesting to elevate the association of inflammatory cytokines with hyperactive platelets or PTEN malfunction in SLE patients. It is also a fruitful field for further research to analyze the status of platelets in patients with autoimmune diseases. Furthermore, the demonstration of inflammatory cytokines or infiltrating platelets in these patients would develop a therapeutic approach to treating such autoimmune diseases as SLE.

It is well established that platelets can extravasate and interact with FRCs around HEV under physiological conditions[52]. The interaction activates platelets via CLEC-2 signal to maintain the HEV integrity[52]. CLEC-2 signal can accelerate platelet activation and aggregate formation[58], providing the potential mechanisms underlying an increase in the number and size of platelet aggregates in T-cell and B-cell zones of lymph nodes in *Pten*^fl/fl^*Pf4-Cre* mice. Of course, collagen, ADP, ATP, and other platelet agonists may contribute to hyper-activation and aggregation of *Pten*-deficient platelets in the lymphoid tissues. Importantly, WT mice generated more platelet aggregates and Tfh cell responses in the lymph nodes after transfer of *Pten*-deficient platelets compared to *Pten*-sufficient platelets. These results support this notion that PTEN expression limits platelets to infiltrate into lymphoid tissues and form aggregates.

In vitro, activated *Pten*-deficient platelets secreted more inflammatory cytokines, including IL-6, IL-17B/F, IL-21, and IFN-γ. Some of these cytokines have been proved to promote T-cell activation, Tfh, and GC B-cell differentiation (Fig. 6h). For example, IL-6 is a potent inducer of Tfh cell differentiation, and IL-6 levels are significantly elevated in serum of SLE patients[18,59]. IL-21 is also important for Tfh and GC B-cell differentiation as well as antibody production[17,51]. Interestingly, PTEN is dispensable for the production and storage of granules and these cytokines. In contrast, PTEN deficiency promoted the secretion of these cytokines by activating the PDK1/mTORC2-AKT-SNAP23 signaling (Fig. 6h). Activation of this signaling induced SNAP23 phosphorylation on Ser95 which has been reported to be crucial for cytokine secretion in platelets[60,61]. Our work identifies PTEN as a negative regulator of SNAP23 phosphorylation and activation. Moreover, *Pten*-deficient platelets were prone to directly interact with CD4^+^ T cells in vitro and in vivo and promote Tfh cell differentiation during coculture. Finally, transfer of PTEN-deficient platelets produced more GC B cells and Tfh cells in pLNs of recipients. Therefore, it is proposed that infiltrating *Pten*-deficient platelets promote aberrant Tfh cell

responses at least in part by direct contact and/or oversecreting inflammatory cytokines. Notably, excess Tfh cells occur in the *Pten*^fl/fl^*Pf4-Cre* mice with normal DCs, suggesting that *Pten*-deficient platelets directly and primarily affect Tfh cell differentiation and contribute to autoimmunity. In addition, *Pten*^fl/fl^*Pf4-Cre* mice can develop excess Tfh cells prior to appearance of autoimmune diseases, aberrant other T helper subsets as well as DCs, suggesting that excess Tfh cells are the most primary contributor to autoimmunity in these mice.

Excess Tfh cell responses result in pathogenic autoantibodies, and is frequently associated with human autoimmune diseases, including SLE[62,63]. Numerous mouse studies have shown a causal function of Tfh cells in autoimmune diseases[63,64]. Human studies also suggest an important action of excess Tfh cells in the pathogenesis of autoimmune diseases[65–67]. Although T-cell-intrinsic mechanisms underlying aberrant Tfh cell differentiation are well understood, the T-cell-extrinsic mechanisms remain unclear[51,68]. Our work identifies platelets as a regulator of Tfh cell differentiation via a cell-extrinsic mechanism.

In addition to excess Tfh cell responses, *Pten*^fl/fl^*Pf4-Cre* mice displayed an increase in activated CD4^+^ T cells and other CD4^+^ helper subsets such as Th1, Th2, and Th17 (Fig. 6h). Some cytokines secreted by platelets have been shown to contribute to the differentiation of these T helper subsets. Besides, platelets have been shown to regulate the activation and effector functions of CD8^+^ T cells, which have been believed to have a function in autoimmunity[69]. Indeed, we observed an increase in activated CD8^+^ T cells in *Pten*^fl/fl^*Pf4-Cre* mice. Collectively, we propose that *Pten*-deficient platelets cause lymphoproliferative and autoimmune diseases through programming multiple types of immune cells.

Although PF4-Cre expression was not limited to platelets, and could be detected in a small population of non-megakaryocytes/platelet hematopoietic lineages including T and B cells. We found that PTEN deficiency in either T and B cells did not confer competitive advantages in *Pten*^fl/fl^*Pf4-Cre* mice. The vast major disease-associated immune cells including GC B cells, Tfh cells, Th1, and active T cells originated from *Pten*-sufficient compartments in these mice, providing evidence that nonspecific PTEN deficiency in either T and B cells has very limited effects on disease progresses. Several studies have shown that PTEN deficiency in myeloid, mast cells, and DCs are able to restrain inflammation, autoimmunity, and Tfh cell responses[43–45]. These results collectively provide evidences that platelet-specific *Pten* deficiency causes the lymphoproliferative and autoimmune diseases in *Pten*^fl/fl^*Pf4-Cre* mice.

## Methods

**Mice.** All experiments were performed in strict accordance with protocols approved by Institutional Animal care and Use Committee (IACUC) of Shanghai Jiao Tong University School of Medicine. All mice were maintained in identical standard conditions (housing, regular care, and normal chow) and in isolated ventilated cages in specific pathogen-free (SPF) facilities. *Pf4* transgenic mice and

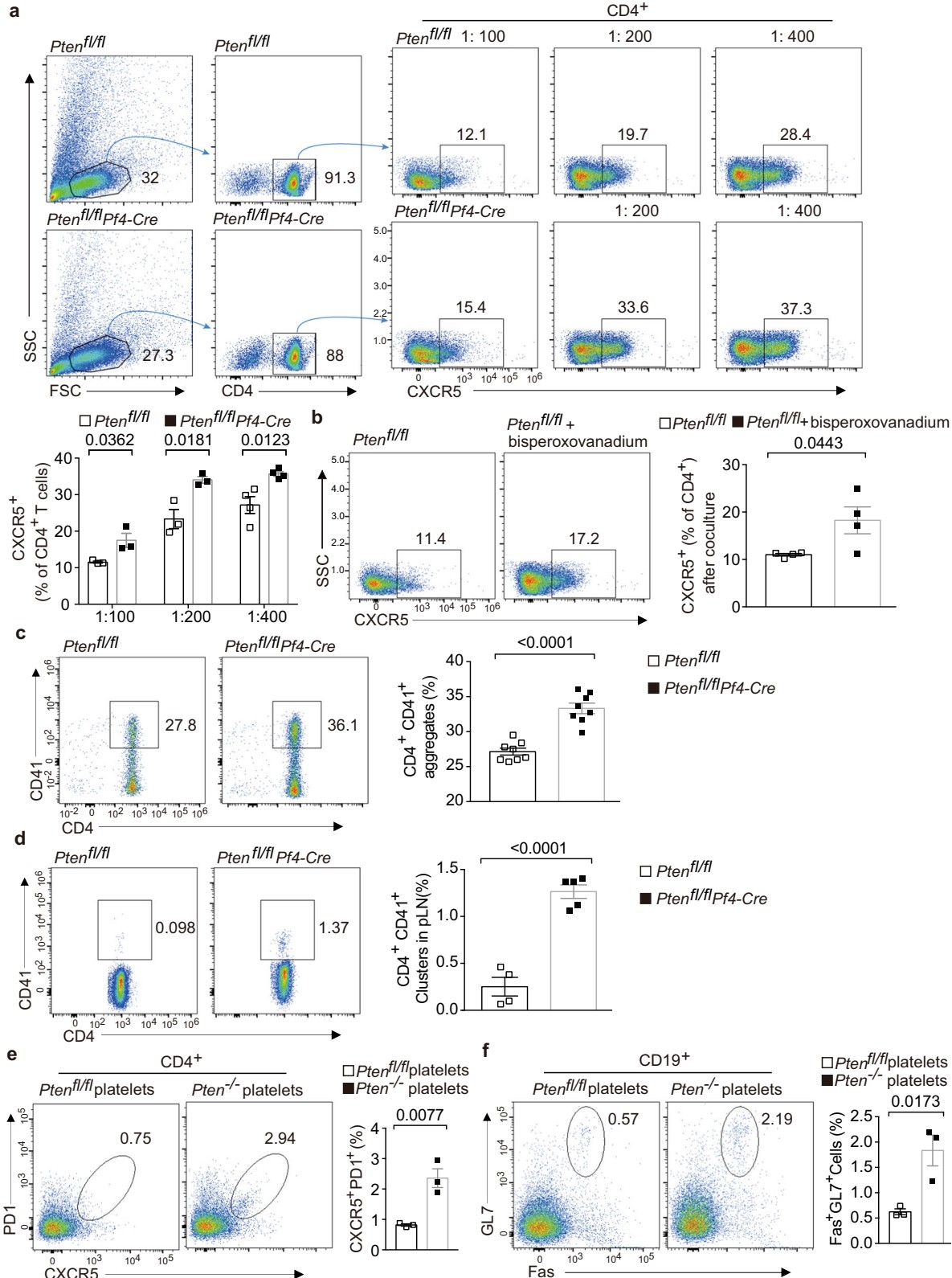

mice with loxP-flanked *Pten* alleles (*Pten*[fl/fl])[38] have been described. *Rosa26*[mT/mG] (Jax007676) mice were from Nanjing Biomedical Research Institute of Nanjing University. The *Rosa26*[mT/mG] reporter mice we used contain a loxP-flanked ('floxed') tandem dimer Tomato gene (mT) and STOP cassette followed by the gene encoding GFP (mG) at the ubiquitously membrane-targeted Rosa26 locus. In this lineage tracing by Cre-Loxp system, PF4-Cre-mediated excision of the floxed STOP cassette results in descendible and constitutive expression of GFP. According to different experimental purposes, these mice were crossed to generate *Pten*[fl/fl]*Pf4-Cre,* *Rosa26*[mT/mG]*Pf4-Cre,* *Rosa26*[mT/mG]*Pten*[fl/fl]*Pf4-Cre,* respectively. All mice were crossed on C57BL/6 background. The development of these lymphoproliferative and autoimmune diseases was monitored by frequent visual examination, flow cytometry, and histopathological analyses. *Pten*[fl/fl]*Pf4-Cre* or *Rosa26*[mT/mG]*Pten*[fl/fl]*Pf4-Cre* mice were analyzed at 3–8 months of age, according to different design in each experiment, and their age- and sex- matched littermate *Pten*[fl/fl] and *Rosa26*[mT/mG]*Pf4-Cre* mice were used as control groups.

**Fig. 5 *Pten*-deficient platelets promote Tfh cell differentiation in vitro and in vivo.** The platelets from 3-month-old mice were subjected for analysis (**a–f**). **a** Flow cytometry analysis and proportions of CXCR5$^+$ populations in CD4$^+$ T cells after coculture with WT and *Pten*-deficient platelets for 4 days (1:100 $n = 3$,1:200 $n = 3$, 1:400 $n = 4$, respectively). **b** Flow cytometry analysis and proportions of CXCR5$^+$ populations in CD4$^+$ T cells after 4-day coculture with *Pten*$^{fl/fl}$ platelets and bisperoxovanadium pretreated WT platelets ($n = 4$). **c** Flow cytometry analysis and quantifications of CD41$^+$ platelet-CD4$^+$ T-cell aggregates in co-cultures with a ratio of 1:200 ($n = 4$). **d** Flow cytometry analysis and quantifications of CD41$^+$ platelet-CD4$^+$ T-cell aggregates in the pLNs of *Pten*$^{fl/fl}$ ($n = 4$) and *Pten*$^{fl/fl}$*Pf4-Cre* mice ($n = 5$). **e, f** Flow cytometry analysis and quantifications of CXCR5$^+$PD1$^+$ Tfh cells and Fas$^+$GL7$^+$ GC B cells in the pLNs from recipients transferred with WT and *Pten*-deficient platelets, respectively ($n = 3$). Results are representative of three independent experiments, and data shown in **a–f** are presented as mean ± s.e.m. (two-tailed *t* test). Source data are provided as a Source Data file.

**Flow cytometry**. Bone marrow cells were flushed from tibias and femurs with sterile FACS buffer (PBS supplemented with 2% FBS and 2 mM EDTA). Splenic and lymphoid cells were gently grinded against the cell strainer by the plunger from a syringe and single-cell suspensions were filtered through fine mesh. In regard to the detection of DCs, splenic tissues were mechanically and enzymatically digested with 1 mg/ml collagenase IV and 50U/ml DNase I for 30 min at 37 °C with rotation. Cells were counted and stained with antibodies in FACS buffer, followed by incubation for 30 min on ice. Staining reagents included fluorescein isothiocyanate (FITC) anti-B220: clone RA3-6B2, allophycocyanin (APC) anti-IgM: clone II/41 (RUO), phycoerythrin (PE) anti-CD24: clone M1/69 (RUO), FITC anti-CD41: clone MWReg30, Percp-Cy5.5 anti-BCL6: clone K112-91, PE-Cy7 anti-CXCR5: clone 2G8, APC anti-IL4: clone 11B11, APC-Cy7 anti-IL17a: clone TC11-18H10, BV421 anti-IFN-γ: clone XMG1.2, FITC anti P-selectin: clone RB40.34, PE-Cy7 anti-Gr-1: clone RB6-8C5, APC anti-Mac1: clone M1/70, PE anti-CD41: clone MWReg30, PE anti-CD138: clone 281-2 were all purchased from BD Biosciences; Pacific Blue anti-CD4: clone GK1.5, Percy-Cy5.5 anti-CD19: clone 6D5, APC anti-CXCR5: clone L138D7, APC anti-CD44: clone IM7, APC-Cy7 anti-CD62L: clone MEL-14, PE-Cy7 anti-CD43: clone 1B11, Pacific Blue anti-GL7: clone GL7, PerCP-Cy5.5 anti-F4/80: clone BM8, APC-Cy7 anti-CD8: clone 53-6.7, PerCP-Cy5.5 Streptavidin were all purchased from Biolegend; APC anti-Fas/CD95: clone 15A7, AF700 anti-Foxp3: clone FJK-16s, APC anti-Cd42d (GPV): clone 1C2, APC anti-CD45: clone 30-F11, APC anti-CD41: clone eBioMWReg30, PE anti-CD62L: clone MEL-14, Biotin AH anti-mouse CD11c: Clone N418, FITC anti-mouse MHC II (I-A/I-E): clone M5/114.15.2, APC Anti-mouse CD86: Clone GL1, PE anti-mouse CD80: clone 16-10A1, PE-Cy7 anti-mouse CD11b: Clone M1/70 were all from eBioscience. Nonspecific binding was blocked with anti-CD16/CD32 (Biolegend): clone 93. Samples were collected on a LSR Fortessa cytometer (BD Biosciences) and then analyzed with FlowJo software. Dead cells and non-singlet events were excluded by 7-amino-actinomycin D staining (eBioscience, 00-6993-50) and the forward scatter area (FSC-A) versus height (FSC-H) characteristics.

**RNA-seq analysis**. Total RNA was harvested from the lymphoid CD4$^+$ T cells and submitted for RNA sequencing using the sequencing platform BGISEQ-500 (BGI). Three biological replicates of each genotype were sequenced. Sequence reads were mapped to mouse genome (mm9) by using bowtie 2 (v 2.3.4.2) with default settings, and reads mapped to multiple positions (MAPQ < 10) were discarded. Gene expression levels were measured by RPKM (reads per kilobase of exon per million reads) by RSEM (v1.2.30). Significantly differentially expressed genes were identified by edgeR (v3.20.9) with the following criteria: Benjamini-Hochberg corrected *P* value < 0.01 and fold-change > 2. Lists of genes differentially expressed by twofold or more were analyzed for functional enrichment using the Ingenuity Pathways Analysis (according to IPA Ingenuity Web Site, www.ingenuity.com. Ingenuity Systems Inc., Redwood City, CA). Differentially expressed genes between CD4$^+$ T cells of pLNs from *Pten*$^{fl/fl}$ and *Pten*$^{fl/fl}$*Pf4-Cre* mice were selected by Limma (v3.42.2). Gene Ontology (GO) Categories enriched analysis was performed by cluster Profiler (v3.14.3). GSEA was performed using the GSEA tool (v4.2.3) (Broad Institute). The gene signature upregulated in Tfh cells relative to their expression in non-Tfh cells were from published data (GEO accession code: GSE21379). The RNA-seq data has been deposited into the GEO series database (GSE143698).

**Histological analysis**. Tissues from *Pten*$^{fl/fl}$*Pf4-Cre* mice and age-matched *Pten*$^{fl/fl}$ littermates (control) were fixed in 4% paraformaldehyde and embedding with paraffine. Sections (4–6 μm in thickness) were cut with a Leica RM2235 Microtome and then examined by Hematoxylin-and-eosin staining. For immunohistochemical analysis, tissue sections were incubated with biotinylated Peanut Agglutinin (PNA, Vector Laboratories, Cat: B-1075) and the signals were detected using VECTAS-TAIN ABC (Avidin-Biotin Complex) peroxidase-based kits from Vector Laboratories (Cat: PK-7100). Immunohistochemistry was performed according to standard procedures.

**Immunostaining**. Lymph nodes were dissected and instantly frozen in Tissue-Tek optimum cutting temperature compound (Sakura, Cat: OCT 4583). Frozen tissues were stored at −80 °C until further processing. Sections (10 μm in thickness) were cut with a Leica Cryostat, mounted on Superfrost Plus glass slides. Tissue cryo-sections were air-dried for 1 h before and after fixation in 4% Paraformaldehyde or

cold acetone for 10 min, and then were rehydrated for 10 min in PBS. Slides were stained overnight at 4 °C in a humidified chamber with the following diluted primary antibodies: anti-B220 (from Biolegend): clone RA3-6B2, PNA (from Vector), anti-CD4 (from Biolegend): clone GK1.5, anti-CD31 (from BD Biosciences): clone MEC13.3, Biotinylated anti-LYVE (from R&D Systems). After washing, secondary antibodies, Alexa Fluor 647 conjugated Goat anti-Rat IgG (Life Technologies), PE-conjugated Goat anti-rat IgG (Rockland, 612-108-120) or APC Streptavidin (BD Biosciences) were applied for 1 h at room temperature. Slides were mounted with the Prolong Gold Antifade reagent (Invitrogen, P36934). GFP-labeled WT (*Rosa26*$^{mT/mG}$*Pf4-Cre*) or *Pten*-deficient (*Rosa26*$^{mT/mG}$ *Pten*$^{fl/fl}$*Pf4-Cre*) platelets were transferred into *Pten*$^{fl/fl}$ mice through tail vein injection. After 2 days, *Pten*$^{fl/fl}$ mice were sacrificed and lymph nodes instantly frozen in Tissue-Tek optimum and stored at −80 °C until further processing or sections were cut as mentioned above.

**Detection of immunoglobulins and anti-dsDNA antibodies**. Immunoglobulin subclasses in plasma were detected by specific ELISA kits (SouthernBiotech 5300-4B) according to the manufacturer's protocol. To detect anti-dsDNA auto-antibodies in plasma, high-binding ELISA plates were coated with 2 μg/ml dsDNA (Sigma-Aldrich, D4522) from calf thymus. Coated plates were blocked, and subsequently incubated with diluted samples overnight at 4 °C in TBS with 1% BSA. Bound anti-dsDNA antibodies were detected with AP-conjugated anti-mouse IgG (Jackson ImmunoResearch, 115-055-146) and streptavidin-HRP (BioTechne, DY998) followed by TMB substrate solution (eBioscience 00-4201-56). Absorbance was measured at 450 nm.

**Platelet preparation, aggregation, and transfer**. The whole blood was collected from abdominal aorta of indicated mice. Platelets were prepared using a centrifuge to separate the platelet-rich plasma from the whole blood, followed by a second centrifugation to concentrate platelets, and then suspended in the Tyrode's buffer ($3 \times 10^8$/ml) for further experiments. Platelet aggregation was measured using the light transmission aggregometry[70]. Platelet purity (>99%) was routinely detected by flow cytometry with platelet-specific CD41 antibody staining. Adenosine 5′-diphosphate (ADP), apyrase, and PGE1 were purchased from Sigma-Aldrich (St. Louis, MO). α-Thrombin was from Enzyme Research Laboratories (South Bend, IN). Collagen was from CHRONO-PAR. Washed *Pten*$^{fl/fl}$, *Pten*$^{fl/fl}$*Pf4-Cre* platelets or *Rosa26*$^{mT/mG}$*Pf4-Cre*, *Rosa26*$^{mT/mG}$*Pten*$^{fl/fl}$*Pf4-Cre* platelets were transferred into *Pten*$^{fl/fl}$ mice via tail vein injection.

**Platelets α-granule secretion**. Washed *Pten*$^{fl/fl}$ and *Pten*$^{fl/fl}$*Pf4-Cre* platelets at a concentration of $3 \times 10^7$/ml were incubated with FITC-labeled P-selectin (BD Biosciences) and 0.1 U/ml of thrombin for 20 min at room temperature in a final volume of 100 μL Tyrode's buffer, and P-selectin binding was measured by flow cytometry. The indicated inhibitors were added to *Pten*$^{fl/fl}$ and *Pten*$^{fl/fl}$*Pf4-Cre* platelets and incubated for 5 min before stimulation with thrombin. AKT inhibitor SH6 and PDK1 Inhibitor II were purchased from Calbiochem. mTORC2 inhibitor PP242 was purchased from Sigma.

**Cytokine protein array**. A RayBio Mouse Cytokine Antibody Array kit (Ray Biotech, Inc. Cat: AAM-CUST-8) was used to detect a panel of 46 secreted cytokines and chemokines that stored in resting platelets or released from activated *Pten*$^{fl/fl}$ and *Pten*$^{fl/fl}$*Pf4-Cre* platelets in response to thrombin, respectively. Prepared platelet purity was routinely detected by flow cytometry with platelet-specific CD41 antibody staining (>99%). The potential contamination of other immune cells in prepared platelets used for these cytokine assays was excluded with leukocyte-specific CD45 antibody staining (<0.05%). The manufacturer's recommended protocol was used to perform this assay. Cytokine protein profiles were quantified by determining band intensities with National Institutes of Health ImageJ software.

**Immunoblot analysis**. Washed platelets prepared and lysed in cell lysis buffer supplemented with Protease and Phosphatase Inhibitor Cocktail (Roche Life Science). Primary antibodies included anti-AKT-Thr308 (Cell Signaling Technology, CST), anti-AKT-Ser473 (CST), anti-SNAP23-Ser95 (GenScript), anti-Sin1-Thr86 (CST), anti-SNAP23 (abnova), anti-PTEN (CST), anti-CD62P

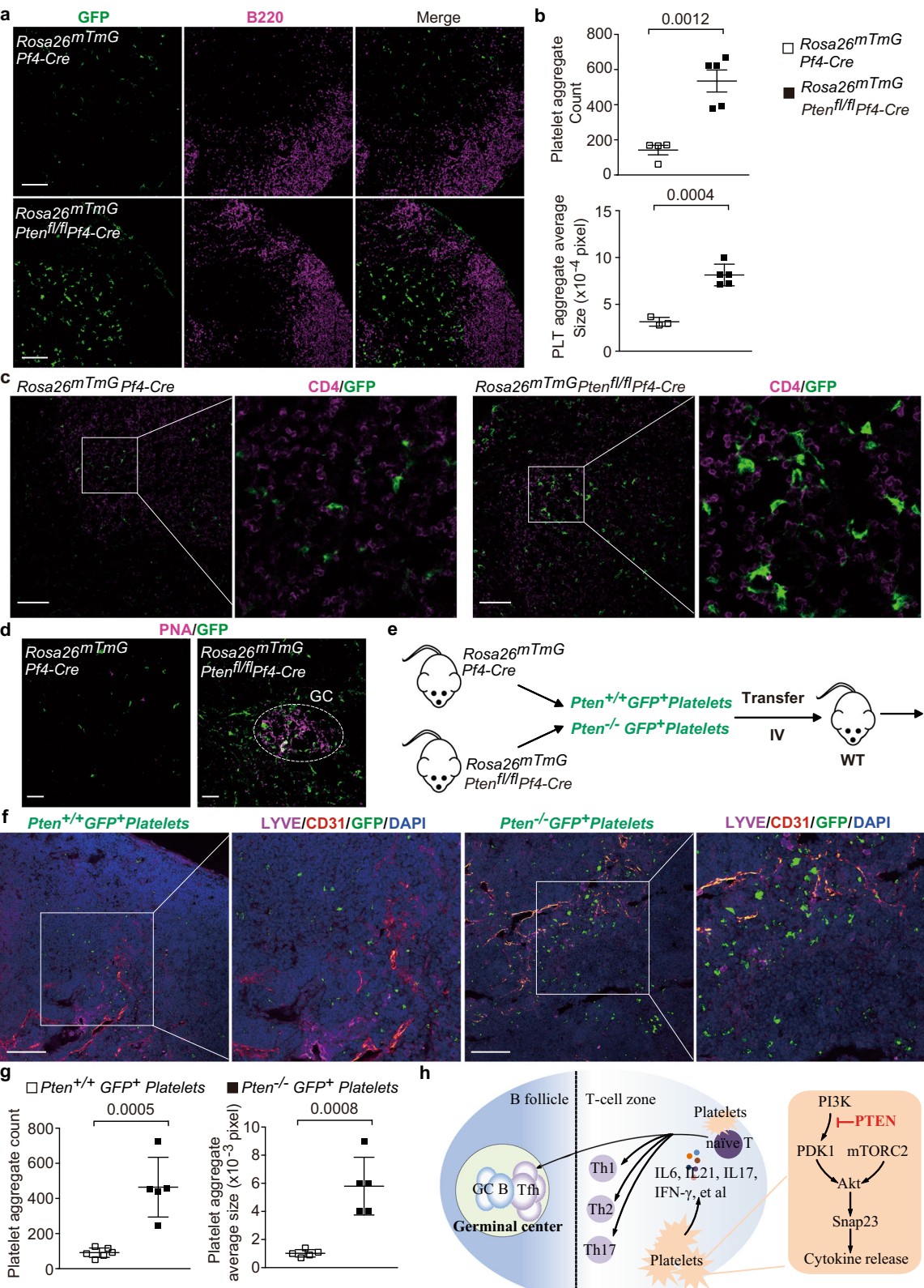

antibody (Abcam) clone: EPR22850-190 and anti-GAPDH (CST) were diluted in 1× TBS buffer (supplemented with 5% BSA or skim milk), followed by incubation at 4 °C overnight.

**Platelet-T-cell coculture.** Washed platelets were prepared as described above. Afterward, the platelets were pelleted and resuspended in DMEM culture medium at the concentration of $1.25 \times 10^9$/ml. The prepared platelets had a leukocyte

contamination rate < 0.05% as assessed by detecting CD45+ events with flow cytometry. CD4+ T cells were isolated using EasySep™ Mouse CD4 negative isolation kit from STEMCELL Technologies (Cat:19852) according to the manufacturer's instructions. The isolated T cells were > 98% CD4 positive and were suspended in DMEM at the cell density of $5 \times 10^6$/ml.

CD4+ T cells were cultured in the absence or presence of $Pten^{fl/fl}$, $Pten^{fl/fl}Pf4$-$Cre$ platelets or PTEN inhibitor bisperoxovanadium (MCE, Cat: HY-122818) pretreated platelets in DMEM containing 5% FBS, 2 μM L-glutamine, 1 μM sodium

**Fig. 6 *Pten*-deficient platelets preferably form aggregates and physically interact with CD4$^+$ T cells in the lymph nodes.** The pLNs from 6-month-old mice were subjected for analysis (**a–d**). The platelets from 3-month-old mice were subjected for analysis (**f**). **a, b** Distribution of platelet aggregates in the pLNs from *Rosa26*$^{mT/mG}$*Pf4-Cre* and *Rosa26*$^{mT/mG}$*Pten*$^{fl/fl}$ *Pf4-Cre* mice (Scale bars, 100 μm). T-cell zone is visualized by B220-negative area (**a**). Quantifications of the number (*Rosa26*$^{mT/mG}$*Pf4-Cre*, n = 4; *Rosa26*$^{mT/mG}$*Pten*$^{fl/fl}$ *Pf4-Cre*, n = 5) and size (*Rosa26*$^{mT/mG}$*Pf4-Cre*, n = 3; *Rosa26*$^{mT/mG}$*Pten*$^{fl/fl}$ *Pf4-Cre*, n = 5) of platelet aggregates in the PLNs (**b**). **c** Direct contact between platelets and CD4$^+$ T cells in T-cell zone of pLNs (Scale bars, 100 μm). **d** The existence of *Pten*-deficient platelets in the germinal centers which identified by PNA-positive clusters and circled by while line (Scale bars, 20 μm). **e** The diagram showing the transfer of GFP-labeled WT (*Pten*$^{+/+}$) and *Pten*-deficient platelets (*Pten*$^{-/-}$) into WT mice. **f** Location of transferred *Pten*$^{+/+}$ GFP$^+$ platelets and *Pten*$^{-/-}$ GFP$^+$ platelets in the lymphoid tissues of recipient mice. The blood vessel and lymphatic vessel are indicated by CD31 positive (red) or LYVE-1positive (purple) region in pLNs of recipient mice. Cell nuclei were counterstained with DAPI (blue) (Scale bars, 100 μm). **g** Quantifications of the number (WT, n = 6; *Pten*-deficient, n = 5) and size (n = 5) of transferred WT and *Pten*-deficient platelet aggregates. **h** An illustration showing aberrant Th and Tfh cell responses caused by *Pten*-deficient platelets and potential underlying mechanisms. Results are representative of at least three independent experiments, and data (**b**, **g**) are presented as mean ± s.e.m. (two-tailed *t* test). Source data are provided as a Source Data file.

pyruvate, 100 U/ml penicillin, and 100 μg/ml streptomycin. CD4$^+$ T-cell activation was induced by coated anti-CD3 mAb (3 μg/ml, 100 μl per well; 4 °C overnight, Invitrogen 16-0031-85) and soluble 0.3 μg/ml anti-CD28 mAb (Invitrogen 16-0281-85). The cell density ratio of T cells to platelets were adjusted to 1:100, 1:200 or 1:400, respectively. T cells and platelets were co-cultured 4 days at 37 °C with 5% CO$_2$. Co-cultured cells were collected and detected for T-cell phenotyping by flow cytometric analysis.

All company names and catalog numbers of antibodies or commercial reagents used in this study were provided in the Supplementary Table 1.

**Statistical analysis**. Data were analyzed with GraphPad Prism 7. Two-tailed unpaired Student's *t* test was used to compare endpoint means of different groups. Experiments were independently repeated at least three times. *P* values less than 0.05 were considered as statistically significant. Error bars on bar graphs represent the standard error of mean (SEM). Total 3–10 mice were used in each experimental group to define meaningful differences between groups. If differences between groups were closed to statistical significance (*P* = 0.1–0.05), more mice were used for a more rigorous test of statistical analysis.

**Reporting summary**. Further information on research design is available in the Nature Research Reporting Summary linked to this article.

## Data availability
The RNA-seq data have been deposited in the Gene Expression Omnibus under accession code: GSE143698. The Tfh cell gene expression data used in this study are available in the GEO database under accession code: GSE21379. All other data are available in the article and its supplementary files or from the corresponding author upon reasonable request. Source data are provided with this paper.

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

## Acknowledgements

This work was supported by National Natural Science Foundation of China (82070119 and 81500106 to X.C., 81525001, 91739302, and 31830050 to J.L., 31870872 to C.H.), and the National Key R&D Program of China (2019YFA0111000 to C.H.). The authors wish to thank Kevin Liu from SHSID for proofreading the article.

## Author contributions

X.C., C.H., and J.L. designed and performed the experiments, analyzed data, and wrote the manuscript; Q.C., H.Z., Y.Z., Y.X., and Y.G. performed some of the experiments; L.S., F.L., L.C., and G.Q.C. helped with the experiments.

## Competing interests

The authors declare no competing interests.
