## [Peer Review File · Nature Communications]

The phosphatase PTEN links platelets with immune regulatory functions of T follicular helper cellsREVIEWER COMMENTS

Reviewer #1 (Remarks to the Author):

Chen and colleagues show that mice with platelet-specific deletion of the phosphate PTEN, develop autoimmunity and lymphoproliferative diseases with age. PTEN deficiency in platelets led to altered T cell differentiation, excessive T follicular helper (Tfh) cells and germinal center (GC) responses. PTEN-deficient were also hyperactive and prone to infiltrate into the T cell zones and GCs of lymph nodes and overproduce Tfh-promoting cytokines, including IL6, IL17B/F, IL21 and IFN- γ , which cause aberrant Tfh cell response and disrupt immune homeostasis. This effect was linked to the activation 15 of PDK1/mTORC2-AKT-SNAP23 pathway in the absence of PTEN. They suggest that platelets have a crucial role in controlling immune homeostasis and that hyperactive platelets act as an important mediators in autoimmunity.

This was an interesting paper, however, there are several points that need to be addressed. In general, the data is largely descriptive and there is little mechanistic experiments to determine how PTEN-deficient platelets are causing all these immune disturbances. For example, Bisperoxovanadium is a selective PTEN inhibitor (see Schmid AC et al. FEBS letters, 2004). Does treatment of WT mice with Bisperoxovanadium cause the same effects as the PTEN-deficiency? Or does it suppress some of the in vitro assays shown?

Throughout the manuscript there are numerous grammatical errors and improper plural/singular usage. The authors need to have the manuscript proof-read by someone competent in English. Page 2/27, line 7. Please define PTEN.

Page 3/27. In the introduction, the authors need to be more comprehensive in the citations related to paragraph 1. I suggest at least adding Kapur R et al., J Immunol 2015 as support for these statements. The authors should also mention that platelets and megakaryocytes are potent antigen presenting cells for CD8+ T cells (see Chapman J Immunol 2012, Zufferey A et al Blood Adv 2018. The introduction requires at least a section on what is already known about how platelets affect both CD4 and CD8 T cells and Th1/Th17 modulation (there is a relatively large literature on this). Furthermore, please introduce PTEN more comprehensively particularly with respect to its role as a tumor suppressor and the role it plays in human cancer and its anti-proliferative effects. Page 8/27, lines 136-143. It is surprising that thrombin stimulation had greater levels of P-selectin on their surface as thrombin is the most potent platelet agonist and all alpha granules are exocytosed in wild type platelets, so the greater amount of P-selectin suggests that there is actually more protein in the PTEN-deficient platelets. This should be shown using quantitative western blotting or SPR.

Page 8/27 lines 149-151. This data is difficult to understand and reconcile. The authors show that IL6, IL17B/F, IL21 and IFN- γ were significantly up-regulated in the supernatant from activated Pten-deficient platelets. Platelets do not make these cytokines nor do they harbor them. What is there source in this assay? There must be contaminating leukocytes in the assays. This needs to be clarified.

Page 9/27, lines 161-169. The authors describe PTEN-deficient platelets migrating into the LN. It is well known that platelets (and RBC) are circulation locked and do not leave the circulation except when there is bleeding. The only lymphoid tissue that platelets (and RBC) interact with is the spleen. Therefore, blood components such as red blood cells and platelets are never seen in lymphatic vessels. Aristotle first recognized this and described lymphatic vessels as "vessels containing a colorless fluid." (see Kim H et al. Circulation Research. 2010;106:1184-1186). This is a very well-known dogma and the authors have not convinced this reviewer that they are actually observing platelets in the nodes. The authors need to bring this out and need to show mechanistic data of how PTEN-deficient platelet actually migrate to the nodes. For example, do PTEN-deficient platelets enter the lymph when they are transfused into WT mice? On the other hand, platelet microvesicles can enter the lymph because of their size and P-selectin content. Are these supposed platelets observed in the LN actually MV?

Page 10/27, lines 179-182. This is pure speculation and should be removed.

Page 11/27, lines 204-209. The authors suggest that the autoimmune phenotypes and aberrant Tfh cell responses described in this study appear not to be caused solely by PTEN deficiency in platelets, however, a small subset of lymphoid lineages also contain PTEN. How do they

differentiate the lymphoid effects from the platelet effects? Where is the data to support that these lymphoid lineages do NOT play a role in the autoimmunity?

Figure 1a. What is the frequency of mice with PTEN-deficient platelets that get lymphoproliferative tumors?

Figure 2 and throughout. The authors show that PTEN deficient platelets cause altered CD4+ T cell differentiation. How do platelets interact with the CD4+ T cells as platelets are absolutely MHC class II negative. Do they do this indirectly via another cell? And if so, how do they do it. What happens to CD8+ T cell differentiation and proliferation? Platelets express MHC class I molecules and can present various peptide antigens to CD8+ T cells. How are these lymphocyte T subsets affected?

Reviewer #2 (Remarks to the Author):

The paper explores how loss of Pten in platelets affects immune phenotype.

They see a very striking lymphoid hyperplasia, activation of T cells and formation of GCs.

Further they show that the platelets in the mice are hyper activated, secrete higher levels of many cytokines and are found in increased numbers within the T cells zones of secondary lymphoid organs compared to wild-type mice.

They conclude that these hyperactivated pten deficient platelets results in increased activation of T and B cells and drive the hyper proliferation.

While these are interesting results I don't believe they have definitively shown that the Pten loss in platelets is the sole cause of the phenotype observed in the mice. As they themselves mention the Pf4-cre is leaky and results in deletion in other cell types such as T and B cells. While they show that there is not expansion of these Pten deleted T and B cells this does not rule out that these cells, a subpopulation of these cells or other cells (e.g. DCs, myeloid cells) are contributing to the phenotype observed in the mice. For example, Pten deletion in Treg has also been shown to lead to hyper proliferation (Shrestha et al 2015) and these were not interrogated in detail here.

This platelet intrinsic nature of the effect could be strengthened by experiments such as:

- Mixed chimeras to demonstrate that the activation of the neutrophils is cell intrinsic
- Repeated transfer of Pten-deficient platelets to WT mice to see if this can induce activation of T and B cells
- More detailed analysis of cre expression in other populations e.g. Tregs, myeloid cells, DCs. In addition the number of replicates in Supp Fig 3d does not give sufficient power to determine that there is no changes e.g. it looks like there may be an increase in GFP+ cells among Tfh cells

Minor points

- Their suggestion that these results are applicable to platelets hyperactivity in general causing increased immune activation, rather than Pten-deficient platelets in particular seems too great a leap. They should be careful not to overinterpret this results.
- Much of their data does not appear to be normally distributed (e.g. Fig 1e), if this is the case a different statistical test rather than a student's t-test should be used
- Why is CXCR5 v Bcl-6 used to define Tfh cells in Fig 2 but CXCR5 v Pd-1 used in suppFig 3? Do the authors have PD-1 staining for the samples in Fig 2 as well?
- Figure 5B – what does the star represent on the lower panel? The presence of a GC?
- Figure 5C – what is the blue staining? It is difficult to tell what is being visualised here.
- Fig 5 - Is there some way to quantitate this data – either quantitate the histology or do flow on the LNs to give the number of platelets
- Is any platelet infiltration seen in the spleen where there is far less hyperplasia?
- Supplementary fig 4a, b – in the CD62L/CD44 plots it looks like there are cells on the x-axis that are being missed.
- Details in methods

- o Please include the clones for Abs
- o How were the platelets prepared for transfer?
- The English could be reviewed to improve readability

Reviewer #3 (Remarks to the Author):

In this manuscript, Chen et al, demonstrated Pten deficiency in platelets resulted in hyperactivation and promoted proinflammatory cytokine production. These cytokines then promoted follicular helper T cell differentiation, which leads to excessive germinal center response and autoimmunity. Aged platelet conditional Pten knockout mice developed spontaneous lupus phenotypes including nephritis. They also explored the regulatory mechanism of Pten in platelets were through the PI3K-AKT pathway. Aberrant platelets are a common clinic feature for multiple autoimmune diseases. These findings are very interesting and could provide some new insights into platelet's regulatory functions affecting immune cells and unappreciated contribution to autoimmune diseases.

Major comments

1. The pro-inflammatory cytokines produced by Pten deficient platelets include IL-6, IL-17, IL-21, IFN-gamma which are critical for Tfh differentiation and function. Besides the effects on Tfh cells, IL-6 is important for Th17 while IL-21, IFN-gamma regulate B cell, CD8+ T differentiation, and function. Pten deficient platelets might promote autoimmunity by mediating multiple effector cells. Authors should show more experimental pieces of evidence for supporting the major conclusions
2. The authors should provide more direct evidence for the hyperactivated platelets promote Tfh response. Whether CD4+ T cells coculture with Pten deficient platelets induce Tfh-signature gene expression?
3. How to exclude the possibility of abnormal innate immune responses caused by Pten deficient platelets contributed to the in vivo autoimmunity?
4. The authors show that Pten deficiency resulted in the upregulation of GC-promoting cytokines. And Pten deficient platelets caused a lupus-like disease in mice. Do SLE patients have PTEN malfunction? Do platelets from SLE patients overproduce these cytokines?
5. How to explain Pten deficient platelets only affected in LN? what attracted the Pten-deficient platelets there? Or Why they can't leave LN but got activated there? What molecules activate platelets in GC?
6. From the data of supplement Fig3 and supplement Fig4, can we draw a conclusion that Pten-KO T or B cells somehow are resistant to the effects caused by Pten deficient platelets?
7. From figure 1e-f, aged platelet conditional Pten knockout mice displayed serological alteration. The author should provide plasma cell phenotype in secondary lymph organs.
8. From figure 3b-c , CD4+ T cells from platelet conditional Pten knockout mice display Tfh-skewing transcriptome. And in figure 4d Pten deficient platelets exhibited elevated cytokines secretion including IL-6, IL-21, and IFN- γ . It is necessary for the author might provide evidence of whether these cytokine signalings are altered in CD4+ T cells in the presence of Pten deficient platelets.

Minor comments

1. In figure 1b, the authors should provide statistical data about splenomegaly and lymphadenopathy.
2. From the data of supplement Fig3 and supplement Fig4, can we draw a conclusion that Pten-KO

T or B cells somehow are resistant to the effects caused by Pten deficient platelets?

3. How does Pten upregulate GC promoting cytokines (the exact mechanism)?

4. The authors show that Pten deficiency resulted in the upregulation of GC promoting cytokines. And Pten deficient platelets caused a lupus-like disease in mice. Do SLE patients have PTEN malfunction? Do platelets from SLE patients overproduce these cytokines?

Point-by-point responses to the reviewer's comments:

We would like to thank the editors and the three reviewers for their positive and encouraging comments as well as their constructive criticisms. We have performed experiments and introduced improvements in the manuscript to address each of these questions. We realize and appreciate that these suggestions have significantly strengthened the manuscript and hence we are truly grateful for the excellent input of the three reviewers. The changes we have made are highlighted by red in the revised manuscript.

To Reviewer #1:

Chen and colleagues show that mice with platelet-specific deletion of the phosphate PTEN, develop autoimmunity and lymphoproliferative diseases with age. PTEN deficiency in platelets led to altered T cell differentiation, excessive T follicular helper (Tfh) cells and germinal center (GC) responses. PTEN-deficient were also hyperactive and prone to infiltrate into the T cell zones and GCs of lymph nodes and overproduce Tfh-promoting cytokines, including IL6, IL17B/F, IL21 and IFN- γ , which cause aberrant Tfh cell response and disrupt immune homeostasis. This effect was linked to the activation of PDK1/mTORC2-AKT-SNAP23 pathway in the absence of PTEN. They suggest that platelets have a crucial role in controlling immune homeostasis and that hyperactive platelets act as an important mediator in autoimmunity.

This was an interesting paper, however, there are several points that need to be addressed.

1. In general, the data is largely descriptive and there is little mechanistic experiments to determine how PTEN-deficient platelets are causing all these immune disturbances. For example, Bisperoxovanadium is a selective PTEN inhibitor (see Schmid AC et al. FEBS letters, 2004). Does treatment of WT mice with Bisperoxovanadium cause the same effects as the PTEN-deficiency? Or does it suppress some of the in vitro assays shown?

Response: We appreciate this helpful comment. Our new co-culture experiments showed that PTEN-deficient platelets increased their ability of interacting with CD4⁺ T cells and promoting them to gain the expression of CXCR5, a hallmark of Tfh cells (revised Fig. 5a). The similar results were observed using PTEN specific inhibitor, bisperoxovanadium (revised Fig. 5b). Mice treated with bisperoxovanadium may disrupt PTEN in multiple types of cells and develop complicated defects. Thus, we respectfully asked to exclude in vivo experiment.

2. Throughout the manuscript there are numerous grammatical errors and improper plural/singular usage. The authors need to have the manuscript proof-read by someone competent in English.

Response: The grammatical errors and improper plural/singular usage have been corrected by a professional English speaker in the revised manuscript.

3. Page 2/27, line 7. Please define PTEN.

Response: Done as suggested (page 2, line 5-6; page 5, line 70-74)

4. Page 3/27. In the introduction, the authors need to be more comprehensive in the citations related to paragraph 1. I suggest at least adding Kapur R et al., J Immunol 2015 as support for these statements. The authors should also mention that platelets and megakaryocytes are potent antigen presenting cells for CD8⁺ T cells (see Chapman J Immunol 2012, Zufferey A et al Blood Adv 2018. The introduction requires at least a section on what is already known about how platelets affect both CD4 and CD8 T cells and Th1/Th17 modulation (there is a relatively large literature on this). Furthermore, please introduce PTEN more comprehensively particularly with respect to its role as a tumor suppressor and the role it plays in human cancer and its anti-proliferative effects.

Response: We thank the reviewer for this important comment. We have carefully revised the introduction section to include the information as suggested by the reviewer (pages 2-4, line 22-57, page 5, line 70-74).

5. Page 8/27, lines 136-143. It is surprising that thrombin stimulation had greater levels of P-selectin on their surface as thrombin is the most potent platelet agonist and all alpha granules are exocytosed in wild type platelets, so the greater amount of P-selectin suggests that there is actually more protein in the PTEN-deficient platelets. This should be shown using quantitative western blotting or SPR.

Response: Western blotting and transmission electron microscopy showed that *Pten*-deficient platelets contained similar levels of total P-selectin and alpha granules to their wild-type counterparts (revised Supplementary Fig. 7a, b). Although thrombin is the most potent platelet agonist, it induces platelets to exocytose alpha granules and express membrane-bound P-selectin in dose-dependent manner (Response Fig. 1). Our results demonstrate that *Pten*-deficient platelets expressed more membrane-bound P-selectin than wild-type platelets after stimulation of lower dose of thrombin (Fig.4b).

Response Fig. 1

Response Fig. 1 Flow cytometric analysis of surface P-selectin on platelets stimulated with indicated doses of thrombin.

6. Page 8/27 lines 149-151. This data is difficult to understand and reconcile. The authors show that IL6, IL17B/F, IL21 and IFN- γ were significantly up-regulated in the supernatant from activated *Pten*-deficient platelets. Platelets do not make these

cytokines nor do they harbor them. What is their source in this assay? There must be contaminating leukocytes in the assays. This needs to be clarified.

Response: We thank the reviewer to clarify this issue. As we stated in the manuscript (page 9, line 170-172), activated platelets can release more than 300 active substances including cytokines, chemokines and growth factors. Flow cytometry was used to exclude the potential contamination of other immune cells in prepared platelets used for cytokine assays. Flow cytometry showed that the percentage of CD45⁺ lymphocytes in prepared platelets was lower than 0.05% (revised Supplementary Fig. 7f). Most importantly, platelet agonist greatly enhanced cytokine production in the supernatant, suggesting that platelets are the resource for cytokines.

7. Page 9/27, lines 161-169. The authors describe PTEN-deficient platelets migrating into the LN. It is well known that platelets (and RBC) are circulation locked and do not leave the circulation except when there is bleeding. The only lymphoid tissue that platelets (and RBC) interact with is the spleen. Therefore, blood components such as red blood cells and platelets are never seen in lymphatic vessels. Aristotle first recognized this and described lymphatic vessels as “vessels containing a colorless fluid.” (see Kim H et al. *Circulation Research*. 2010;106:1184–1186). This is a very well-known dogma and the authors have not convinced this reviewer that they are actually observing platelets in the nodes. The authors need to bring this out and need to show mechanistic data of how PTEN-deficient platelet actually migrate to the nodes. For example, do PTEN-deficient platelets enter the lymph when they are transfused into WT mice? On the other hand, platelet microvesicles can enter the lymph because of their size and P-selectin content. Are these supposed platelets observed in the LN actually MV?

Response: We thank the reviewer for raising this important issue. It is well-known that T/B lymphocytes can enter lymph nodes via high endothelial venule (HEV). In fact, platelets can extravasate and interact with fibroblastic reticular cells (FRCs) around HEV (Page 12, line 221-230 and page 13, line 254-258) (see Herzog BH, *et al. Nature* **502**, 2013, 105-109). The transmembrane O-glycoprotein Podoplanin (PDPN),

expressed on FRCs, acts as an activating ligand for platelet C-type lectin-like receptor 2 (CLEC-2). Activated platelets release sphingosine-1-phosphate and further promote VE-cadherin expression on HEVs, maintaining HEV integrity in lymph nodes. Our new experiments showed that transferred platelets can extravasate and form aggregates in the lymph nodes (revised Fig. 6f). *Pten*-deficient platelets formed a greater number of aggregates with larger size in the lymph nodes of recipient mice than wild-type platelets (revised Fig. 6f, g). Interestingly, upon CLEC-2 ligation, PTEN deficiency accelerated the formation of platelet aggregates in vitro (Response Fig. 2). Accumulating evidences have shown that platelets can infiltrate into tissues under inflammation conditions (see Kapur R, Semple JW. *J Exp Med* 218, 2021; Rossaint J, *et al. J Exp Med* 218, 2021; Malehmir M, *et al. Nat Med* **25**, 2019, 641-655). We reason that extravasated *Pten*-deficient platelets prefer to form aggregates in lymph nodes, probably via activation of CLEC-2 signaling. This notion has been included in the discussion section (page 13, line 254-258). We are continuing to work on this signaling as we also think it is important for understanding the mechanism underlying *Pten*-deficient platelets extravasated into lymph nodes and got activated there, and eventually led to autoimmunity. Of course, other platelet agonists such as collagen and ADP exist in the lymphoid nodes, contributing to formation of *Pten*-deficient platelet aggregates. Finally, platelet microvesicles usually carry inflammatory cytokines and might shed from extravasated activated platelets to spread inflammation through whole lymphoid tissues.

Moreover, GFP⁺ platelets were repeatedly detected in the lymphoid section of report *Rosa26^{mTmG}Pf4-Cre* mice (revised Fig. 6 a-d), as well as recipient mice transferred with GFP⁺ platelets (revised Fig. 6 f, g). Meanwhile, we detected more *Pten*-deficient GFP⁺ platelets extravasated into lymph nodes and physically contact with CD4 T cells in these mice either by confocal or flow cytometry (revised Fig. 6 c and revised Fig. 5d). The statistical analysis confirmed *Pten*-deficient GFP⁺ platelets displayed a significantly increased number and size in lymph nodes of these mice (revised Fig. 6b, g), indicating that *Pten*-deficient platelets were enhance their ability of entrance into

the lymph nodes. We believe these new results could convince reviewer and strengthen the conclusion that we are actually observing platelets in the lymph nodes.

Response Fig. 2

Response Fig. 2 PTEN deficiency enhanced CLEC2-mediated platelet activation. The formation of aggregates for WT and *Pten*-deficient platelet in response to 2 μ g/mL CLEC-2 mAb (monoclonal antibody) and Rat IgG isotype control. Scale bar showed the time of platelet aggregate formation.

8. Page 10/27, lines 179-182. This is pure speculation and should be removed.

Response: Done as suggested.

9. Page 11/27, lines 204-209. The authors suggest that the autoimmune phenotypes and aberrant Tfh cell responses described in this study appear not to be caused solely by PTEN deficiency in platelets, however, a small subset of lymphoid lineages also contain PTEN. How do they differentiate the lymphoid effects from the platelet effects? Where is the data to support that these lymphoid lineages do NOT play a role in the autoimmunity?

Response: As the reviewer mentioned, PF4-Cre mediated *Pten* deletion is not limited to platelets. Indeed, PF4-Cre expression was detected in multiple lineages. However, PF4-Cre mediated *Pten* deletion seemed to occur in very small populations of CD4⁺ T and CD19⁺ B lineages (less than 3%, revised Supplementary Fig. 4), and these defects did not lead to competitive advantage (less than 5%, revised Supplementary Fig. 4). Importantly, more than 95% disease-associated T or B subsets including Tfh and GC B cells originated from *Pten*-sufficient T or B cells in *Pten^{fl/fl} Pf4-Cre* mice

(Supplementary Fig. 5 and 6). In addition, *Pten* deficiency in myeloid, mast cells and DCs was reported to restrain inflammation, autoimmunity and Tfh cell responses (page 7-8, line 130-135). These results collectively provide evidences that platelet-specific *Pten* deficiency mainly cause the lymphoproliferative and autoimmune diseases in *Pten^{fl/fl}Pf4-Cre* mice. This notion has been mentioned in the discussion section (page 15, line 295-304).

10. Figure 1a. What is the frequency of mice with PTEN-deficient platelets that get lymphoproliferative tumors?

Response: All *Pten^{fl/fl}Pf4-Cre* mice were normal in appearance at the age of 4-5 weeks, 58 out of 66 these mice (87.9%) eventually developed lymphoproliferative diseases within two years (revised Fig. 1h).

11. Figure 2 and throughout. The authors show that PTEN deficient platelets cause altered CD4⁺ T cell differentiation. How do platelets interact with the CD4⁺ T cells as platelets are absolutely MHC class II negative. Do they do this indirectly via another cell? And if so, how do they do it. What happens to CD8⁺ T cell differentiation and proliferation? Platelets express MHC class I molecules and can present various peptide antigens to CD8⁺ T cells. How are these lymphocyte T subsets affected?

Response: We thank the reviewer for raising the important points. Platelets have been shown to influence the activation and differentiation of immune cells including CD4⁺ and CD8⁺ T cells via physical interaction and/or secreting multiple types of inflammation cytokines (page 2-4, line 19-57). The physical interaction is mediated by a variety of receptors including platelet P-selectin, CD40/CD40L and GPIb, as well as their counter receptors PSGL-1, CD40L/CD40 and Mac-1 on other immune cells (page 3-4, line 37-57). In addition, platelets and megakaryocytes have been reported to process and present both foreign and self-antigens to CD8⁺ T cells via their expressing of MHC class I molecules (page 2-3, line 22-24). Our new

experiments revealed aberrant activation of CD8⁺ T cells in *Pten^{fl/fl}Pf4-Cre* mice (revised Fig. 2b, Supplementary Fig. 3b).

To Reviewer #2:

The paper explores how loss of Pten in platelets affects immune phenotype. They see a very striking lymphoid hyperplasia, activation of T cells and formation of GCs.

Further they show that the platelets in the mice are hyper activated, secrete higher levels of many cytokines and are found in increased numbers within the T cells zones of secondary lymphoid organs compared to wild-type mice.

They conclude that these hyperactivated pten deficient platelets results in increased activation of T and B cells and drive the hyper proliferation.

While these are interesting results I don't believe they have definitively shown that the Pten loss in platelets is the sole cause of the phenotype observed in the mice. As they themselves mention the Pf4-cre is leaky and results in deletion in other cell types such as T and B cells. While they show that there is not expansion of these Pten deleted T and B cells this does not rule out that these cells, a subpopulation of these cells or other cells (e.g. DCs, myeloid cells) are contributing to the phenotype observed in the mice. For example, Pten deletion in Treg has also been shown to lead to hyper proliferation (Shrestha et al 2015) and these were not interrogated in detail here.

This platelet intrinsic nature of the effect could be strengthened by experiments such as:

- Mixed chimeras to demonstrate that the activation of the neutrophils is cell intrinsic

Response: We agree with the reviewer that it is valuable to examine whether the activation of lymphocytes is cell-intrinsic. To address this question, we used *Rosa26^{mT/mG}Pten^{fl/fl}Pf4-Cre* mice in which *Pten*-deficient and wild-type compartments co-exist. PTEN deletion in CD4⁺ T and CD19⁺ B lineages did not

confer competitive advantage in activated T cells, Th1, GC B cells and Tfh cells in *Rosa26^{mT/mG}Pten^{fl/fl}Pf4-Cre* mice (Supplementary Fig. 5). However, *Pten*-sufficient compartment developed excess of activated T cells, Th1, Tfh cells and GC B cells in *Rosa26^{mT/mG}Pten^{fl/fl}Pf4-Cre* mice, but not *Rosa26^{mT/mG}Pf4-Cre* mice (Supplementary Fig. 6). These results demonstrate the defects of lymphocytes are cell-extrinsic in *Pten^{fl/fl}Pf4-Cre* mice.

- Repeated transfer of *Pten*-deficient platelets to WT mice to see if this can induce activation of T and B cells

Response: Wild-type mice were transferred wild-type or *Pten*-deficient platelets weekly via tail vein injection. We found that mice receiving *Pten*-deficient platelets developed excess of GC B cells and Tfh cells on week 6 after transfer (revised Fig. 5e, f), although they have not developed lymphoproliferative and autoimmune diseases at this time-point. In fact, all *Pten^{fl/fl}Pf4-Cre* mice at the age of 4-6 week were normal in appearance, suggesting that *Pten*-deficient platelets-mediated these defects are long-term processes. Unfortunately, mice receiving *Pten*-deficient platelets eventually died around 8 weeks after transfer due to overproduction of thrombus, hindering us to analyze the long-term effect of *Pten*-deficient platelets.

- More detailed analysis of cre expression in other populations e.g. Tregs, myeloid cells, DCs. In addition the number of replicates in Supp Fig 3d does not give sufficient power to determine that there is no changes e.g. it looks like there may be an increase in GFP+ cells among Tfh cells

Response: We thank the reviewer for these important comments. In addition to platelets, PF4-Cre mediated-*Pten* deletion occurred in other lineages, probably contributing to the autoimmune diseases. Less than 3% of CD4⁺ T and CD19⁺ B lineages expressed PF4-Cre (revised Supplementary Fig. 4). Our new experiments further revealed that all Treg cells were negative for PF4-Cre (revised Supplementary Fig. 4). PF4-Cre-mediated *Pten* deletion in CD4⁺ T and CD19⁺ B lineages did not

confer competitive advantage in disease-associated immune cells including Tfh cells (Supplementary Fig. 5). Compared to T/B cells, more myeloid lineages expressed PF4-Cre (revised Supplementary Fig. 4), and this tendency was increased in the absence of *Pten*. However, *Pten* deficiency in myeloid, mast cells and DCs was shown to restrain inflammation, autoimmunity and Tfh cell responses (page 7-8, line 130-135. page 15, line 295-304). Therefore, we propose that PTEN deficiency in platelets mainly leads to autoimmune diseases in *Pten^{fl/fl}Pf4-Cre* mice.

Minor points

1. - Their suggestion that these results are applicable to platelets hyperactivity in general causing increased immune activation, rather than Pten-deficient platelets in particular seems too great a leap. They should be careful not to overinterpret this results.

Response: We have made the changes as suggested by the reviewer.

2. - Much of their data does not appear to be normally distributed (e.g. Fig 1e), if this is the case a different statistical test rather than a student's t-test should be used

Response: Thanks for your advice. Multiple statistical tests were included in this study and presented at the end of legends for each figure.

3. - Why is CXCR5 v Bcl-6 used to define Tfh cells in Fig 2 but CXCR5 v Pd-1 used in suppFig 3? Do the authors have PD-1 staining for the samples in Fig 2 as well?

Response: Tfh cells are identified as CXCR5⁺ PD-1⁺ or CXCR5⁺ Bcl6⁺ population. In most cases, we used two different methods to define Tfh cells. We have shown CXCR5⁺ PD-1⁺ Tfh cells in the revised Fig. 2d as suggested.

4. - Figure 5B – what does the star represent on the lower panel? The presence of a GC?

Response: I am sorry for this carelessness. PNA-positive area represents GC, which is circled in the revised Fig. 6d. The related interpretation has been added in the figure legend (revised Fig. 6d).

5. - Figure 5C – what is the blue staining? It is difficult to tell what is being visualised here.

Response: DAPI was used to visualize nuclei, and its color was blue. We have stated this at the figure legend of revised Fig. 5f.

6. - Fig 5 - Is there some way to quantitate this data – either quantitate the histology or do flow on the LNs to give the number of platelets

Response: Quantifications of the number and average size of platelet aggregates in pLNs have been included (revised Fig. 6b, g). Moreover, the flow cytometry plots showing the proportions of platelets-binding CD4⁺ T cells and quantifications were added (revised Fig. 5d).

7. - Is any platelet infiltration seen in the spleen where there is far less hyperplasia?

Response: The platelet infiltration in various lymphoid tissues varied in different lymph nodes and spleen. Platelet aggregates can form in the white pups of the spleen of 6-month-old *Rosa26^{mT/mG}Pten^{fl/fl}Pf4-Cre* mice (Response Fig. 3).

Response Fig. 3

Response Fig. 3 Increased platelet infiltration in the white pulp of the spleen from 6-month-old *Rosa26^{mT/mG}Pten^{fl/fl}Pf4-Cre* mice.

8. - Supplementary fig 4a, b – in the CD62L/CD44 plots it looks like there are cells on the x-axis that are being missed.

Response: We have modified the flow cytometry plots to avoid this situation (revised Supplementary Fig. 6a, b).

9. Details in methods. Please include the clones for Abs

Response: Done as suggested (page 16, 17, 18).

o How were the platelets prepared for transfer?

Response: The method has been included (page 19, line 390-392, line 394-396).

10. - The English could be reviewed to improve readability

Response: Done as suggested.

To reviewer #3:

In this manuscript, Chen et al, demonstrated Pten deficiency in platelets resulted in hyperactivation and promoted proinflammatory cytokine production. These cytokines then promoted follicular helper T cell differentiation, which leads to excessive germinal center response and autoimmunity. Aged platelet conditional Pten knockout mice developed spontaneous lupus phenotypes including nephritis. They also explored the regulatory mechanism of Pten in platelets were through the PI3K-AKT pathway. Aberrant platelets are a common clinic feature for multiple autoimmune diseases. These findings are very interesting and could provide some new insights into platelet's regulatory functions affecting immune cells and unappreciated contribution to autoimmune diseases.

Major comments

1. The pro-inflammatory cytokines produced by *Pten* deficient platelets include IL-6, IL-17, IL-21, IFN-gamma which are critical for Tfh differentiation and function. Besides the effects on Tfh cells, IL-6 is important for Th17 while IL-21, IFN-gamma regulate B cell, CD8⁺ T differentiation, and function. *Pten* deficient platelets might promote autoimmunity by mediating multiple effector cells. Authors should show more experimental pieces of evidence for supporting the major conclusions

Response: We thank the reviewer for raising the important point. In addition to Tfh cells, *Pten*^{fl/fl}*Pf4-Cre* mice displayed aberrant phenotypes in multiple immunes including Th1, Th2, Th17 and CD8⁺ T cells (revised Fig. 2 and Supplementary Fig. 3). Thus, it is possible that *Pten*-deficient platelets caused autoimmunity via multiple effector immune cells. This notion has been emphasized in the revised manuscript (page 14-15, line 286-294). The aim of this study is to establish the crucial role of PTEN in platelets for maintenance of immune homeostasis and explore the potential cellular mechanisms. Moreover, it was shown that *Pten*-deficient platelets were prone to directly interact with CD4⁺ T cells and promote Tfh cell differentiation both in vitro and in vivo (revised Fig. 5 and revised Fig. 6c). Excessive Tfh cell responses result in pathogenic autoantibodies. Numerous mouse studies have revealed a causal role of Tfh cells in autoimmune diseases. Human studies also suggest an important role of excess Tfh cells in the pathogenesis of autoimmune diseases. Recently Tfh cells have received extensive attentions due to the importance in human autoimmunity (page 14, line 274-285). Our new findings demonstrated that platelets act as a new regulator of Tfh cell differentiation via a cell-extrinsic mechanism. Considering that the effects of platelets on the differentiation of other CD4⁺ T helper lineages have been established, we focus on Tfh cells in this study.

3. How to exclude the possibility of abnormal innate immune responses caused by *Pten* deficient platelets contributed to the in vivo autoimmunity?

Response: This is a very interesting question. Indeed, platelets can influence the differentiation and functions of innate immune cells including DCs. We agree with

the reviewer that *Pten*-deficient platelets might influence innate immune cells to promote autoimmunity. The aim of this study is to establish the crucial role of PTEN in platelets in maintenance of immune homeostasis and explore the potential cellular mechanisms. We mainly focus on Tfh cells in this study since excess Tfh cell responses act as crucial mediators in the pathogenesis of human and mouse autoimmune diseases. As we stated in the revised manuscript that *Pten*-deficient platelets caused autoimmunity via multiple adaptive and innate immune cells (page 14-15, line 286-294). We respectively asked to exclude the study suggested by the reviewer.

4. The authors show that *Pten* deficiency resulted in the upregulation of GC-promoting cytokines. And *Pten* deficient platelets caused a lupus-like disease in mice. Do SLE patients have PTEN malfunction? Do platelets from SLE patients overproduce these cytokines?

Response: Due to time limitation, we only analyzed three SLE and three normal cases during we revised this manuscript. Interestingly, platelets from one SLE patient overproduced IFN- γ , a driven inflammatory cytokine for SLE (Response Fig. 4). Immunoblot analysis showed that the protein abundances of PTEN were comparable between SLE and normal controls (Response Fig. 4). We cannot exclude the possibility that the activity of PTEN was altered in platelets from SLE since abnormal PTEN post-modifications have been reported to impair PTEN function (see Kathryn L. Post, *et al. Nature Communications*, 2020; Justin Taylor & Omar Abdel-Wahab, *nature cell biology*, 2019; X Wang and X Jiang *et al. Oncogene*, 2008). In addition, more SLE cases should be included to draw a conclusion. We are continuing to work on this as we also think this is important for understanding the mechanism underlying *Pten*-deficient platelets-mediated autoimmunity.

Response Fig. 4 IFN γ secretion and PTEN protein expression in platelets. a Representative flow cytometry plots of IFN- γ produced by platelets from Healthy control and SLE patient. **b** Detection of PTEN protein levels in platelets of Healthy controls and SLE patients.

5. How to explain Pten deficient platelets only affected in LN? what attracted the Pten-deficient platelets there? Or Why they can't leave LN but got activated there? What molecules activate platelets in GC?

Response: We thank the reviewer for these important questions. Indeed, *Pten*-deficient platelets can infiltrate into many lymphoid tissues including spleen (Response Fig. 3). The differential microelement between spleen and lymphoid nodes may explain diverse outcomes. In fact, we observed that lymphoid lymphoproliferation also varied across different lymphoid nodes. Platelets can extravasate and interact with fibroblastic reticular cells (FRCs) around HEV in lymph nodes (Page 12, line 221-230 and page 13, line 254-258). The transmembrane O-glycoprotein Podoplanin (PDPN), expressed on FRCs, acts as an activating ligand for platelet C-type lectin-like receptor 2 (CLEC-2). Interestingly, upon CLEC-2 ligation, PTEN deficiency accelerates the platelet aggregation in vitro (Response Fig. 2). We reason that extravasated PTEN-deficient platelets prefer to form aggregates in lymphoid tissues, probably via CLEC-2 signaling. We are planning to work on this signaling as we also think this is important for understanding the mechanism underlying *Pten*-deficient platelets extravasated into lymph nodes and got activated there, and eventually led to autoimmunity. This CLEC-2-PDPN axis mediated platelet

extravasation and activation maintains HEV integrity via S1P release and further VE-cadherin expression (see Herzog BH, *et al. Nature* **502**, 2013, 105-109). Moreover, the released S1P, a bioactive metabolite of sphingosine, plays a pivotal intrinsic role in platelet activation and can further activate platelets around. Of course, other platelet agonists such as collagen and ADP exit in the lymphoid nodes, contributing to the formation of platelet aggregates (page 13, line 259-260). Collagen, ADP or other unknown platelet agonists locate to GC to activate platelets. It is also possible that platelets have been activated before moving into GC.

Response Fig. 2

Response Fig. 2 PTEN deficiency enhanced CLEC2-mediated platelet activation. The formation of aggregates for WT and *Pten*-deficient platelet in response to 2 μ g/mL Clec2 mAb (monoclonal antibody) and Rat IgG isotype control. Scale bar showed the time of platelet aggregate formation.

Response Fig. 3

Response Fig. 3 Increased platelet infiltration in the white pulp of the spleen from 6-month-old Rosa26^{mTmG} *Pten^{fl/fl}* Pf4-Cre mice.

6. From the data of supplement Fig3 and supplement Fig4, can we draw a conclusion

that *Pten*-KO T or B cells somehow are resistant to the effects caused by *Pten* deficient platelets?

Response: Although PTEN loss intrinsically promotes the expansion of B and T cells, we found that PTEN deletion in T and B lineages did not confer competitive advantage in activated T cells, Th1, GC B cells and Tfh cells in *Pten^{fl/fl}Pf4-Cre* mice (Supplementary Fig. 5-6). It is likely that *Pten*-deficient platelets had dominant effects in the autoimmune diseases, and the status of PTEN in these small subsets of T and B cells was neglectful at the presence of *Pten*-deficient platelets.

In addition, as you mentioned that the expansion of activated, Th1 and Tfh subsets as well as GC B cells occurred in GFP⁻ *Pten*-sufficient T or B cells rather than GFP⁺ *Pten*-deficient T or B cells (Supplementary Fig. 5 and Supplementary Fig. 6), indicating *Pten*-deficient T or B cells might resistant to the effects caused by *Pten*-deficient platelets. It has been shown that PTEN deficiency in myeloid cells has resistant effects in autoimmunity, which raises the interesting question whether *Pten*-deficient T or B cells are also resistant to the autoimmunity caused by *Pten*-deficient platelets. However, expansion of GFP⁻ *Pten*-sufficient rather than GFP⁺ *Pten*-deficient compartments might due to *Pten*-sufficient compartments have more opportunities to contact with platelets because of their huge numbers relative to PTEN-deficient compartments. This question is interesting but beyond the scope of this study, and further work is required to elucidate it.

7. From figure 1e-f, aged platelet conditional *Pten* knockout mice displayed serological alteration. The author should provide plasma cell phenotype in secondary lymph organs.

Response: It is a great comment. We found significantly increased CD138⁺ plasma cells in both pLNs (revised Fig. 2g) and spleen (revised Supplementary Fig. 3g) of *Pten^{fl/fl}Pf4-Cre* mice as compared with littermate controls.

8. From figure 3b-c, CD4⁺ T cells from platelet conditional *Pten* knockout mice

display Tfh-skewing transcriptome. And in figure 4d Pten deficient platelets exhibited elevated cytokines secretion including IL-6, IL-21, and IFN- γ . It is necessary for the author might provide evidence of whether these cytokine signalings are altered in CD4⁺ T cells in the presence of Pten deficient platelets.

Response: Gene ontology and Kyoto Encyclopedia of Gene and Genomes (KEGG) pathway analysis demonstrated that the gene signatures activated by IL-6, IL-17 or IFN- γ were significantly overrepresented in CD4⁺ T cells isolated from *Pten*^{fl/fl}*Pf4-Cre* mice (revised Fig. 4e).

Minor comments:

1. In figure 1b, the authors should provide statistical data about splenomegaly and lymphadenopathy.

Response: Done as suggested (revised Fig. 1h, revised Supplementary Fig. 1c).

2. From the data of supplement Fig3 and supplement Fig4, can we draw a conclusion that Pten-KO T or B cells somehow are resistant to the effects caused by Pten deficient platelets?

Response: The response to this comment has been mentioned in major comment 6.

3. How does Pten upregulate GC promoting cytokines (the exact mechanism)?

Response: *Pten*-deficient platelets over-secrete multiple GC-promoting cytokines including IL-6, IL-21 and IFN- γ (Fig. 4d). PTEN deficiency upregulated AKT, PDK1 and mTORC2 pathways, subsequently inducing excess SNAP23 phosphorylation (Fig. 4 f). SNAP23 phosphorylation enhanced the secretion of cytokines stored in granules, including IL-6, IL-21 and IFN- γ (Fig. 4 b, g). We have illustrated a working model depicting the mechanism underlying PTEN-dependent secretion of cytokines (Fig. 6h), and the related information has also been included in the discussion section (page 14, line 270-273).

4. The authors show that Pten deficiency resulted in the upregulation of GC promoting cytokines. And Pten deficient platelets caused a lupus-like disease in mice. Do SLE patients have PTEN malfunction? Do platelets from SLE patients overproduce these cytokines?

Response: The response to this comment has been mentioned in major comments 4.

REVIEWER COMMENTS

Reviewer #1 (Remarks to the Author):

The authors have adequately addressed all of my comments.

Reviewer #2 (Remarks to the Author):

The authors have addressed my concerns. I have no further comments

Reviewer #3 (Remarks to the Author):

The authors have tried to address most of the concerns raised by the reviewers. I still have some major concerns about this revision.

1. I don't think the revised title "Platelets require the phosphatase PTEN to maintain immune homeostasis and restrain Tfh cell responses". Platelets could affect immune response but there is no solid data to support this strong statement.
2. The authors still have not presented convincing data regarding the intrinsic mechanisms for PTEN deficient platelets mediated autoimmunity.
3. PTEN deficient platelets can cause autoimmune phenotypes are novel findings but It is not clear which immune cell compartment(s) are a primary contributor. Excessive Tfh activation could be a secondary effect. More comprehensive studies are needed to be done as the reviewers suggested.
3. The disease relevance of this study is weak. More data from suitable patients or adding more data from autoimmune diseases models would be informative.

Point-by-point responses to the reviewer's comments:

We would like to thank the editors and reviewer for their positive and encouraging comments as well as their constructive criticisms. The changes we have made are highlighted by red in the revised manuscript.

To Reviewer #3:

The authors have tried to address most of the concerns raised by the reviewers.

I still have some major concerns about this revision.

1. I don't think the revised title "Platelets require the phosphatase PTEN to maintain immune homeostasis and restrain Tfh cell responses". Platelets could affect immune response but there is no solid data to support this strong statement.

Response: *Pten^{fl/fl}Pf4-Cre* mice developed aberrant Tfh cell responses and aged-related autoimmunity. *Pf4-Cre* is generally believed to be platelet-specific *Cre* for decades (see Ralph Tiedt *et al*, *Blood*, 2007) and has been extensively used to effectively delete targeted gene in megakaryocytes/platelets in thousands of literatures. Although *Pf4-Cre*-mediated recombination might occur in some small population of non-megakaryocyte cells under some conditions, our study revealed that *Pten*-deficient platelets are major driver of autoimmunity in *Pten^{fl/fl}Pf4-Cre* mice.

Of course, other immune cells educated by *Pten*-deficient platelets, such as Tfh, T helper, CD8⁺ T cells, monocytes, DCs are predicted to contribute to the autoimmune diseases in these mice. However, it has been established that platelet can educate T helper, CD8⁺ T cells, macrophages/monocytes or DCs and play important roles in many autoimmune, inflammatory or infectious diseases (see Starossom SC, *et al. Circ Research* 2015; Elzey BD, *et al. Blood*. 2008; Iannacone M, *et al. Nat Med*. 2005; Mantovani A, Garlanda C. *Nat Immunol* 2013; Hottz ED, *et al. Blood* 2020; Li Guo, *et al. Blood*. 2021; Rossaint J, *et al. J Exp Med* 2021; Linge P, *et al. Nat Rev Rheumatol* 2018; Qiu J, *et al. Sci Rep* 2016; Han P, *et al. Science Advance* 2020; Duffau P, *et al. Sci Transl Med* 2010). Thus, we focus on Tfh cells in this study.

We agree with the reviewer that the title seems to neglect the contributions of other immune cells especially those affected by *Pten* deficient platelets in *Pten^{fl/fl}Pf4-Cre*

mice. The revised manuscript is entitled “The phosphatase PTEN links platelets with immune regulatory functions”.

2. The authors still have not presented convincing data regarding the intrinsic mechanisms for PTEN deficient platelets mediated autoimmunity.

Response: Our study demonstrated that *Pten* deficient platelets enhanced the ability of infiltration into the lymphoid tissues (revised Fig. 6a-g), interaction with CD4⁺ T cells (revised Fig. 5c, d and Fig. 6c), further promoting Tfh-signature gene expression (revised Fig. 5a-f), and secreted more inflammatory and Tfh promoting cytokines (Fig. 4d, e). *Pten* deficient platelets significantly increased expression of CXCR5, a hallmark of Tfh cells, both under in vitro co-culture (revised Fig. 5a) and in vivo platelet transferred conditions (revised Fig. 5e, f). Excessive Tfh cell formation is responsible for autoimmunity (see Joseph E. Craft, *Nat Rev Rheumatol.* 2012; Fan Xiao, *et al. cellular & molecular immunology.* 2021; Roza I Nurieva & Yeonseok Chung, *cellular & molecular immunology.* 2012). Tfh cells are recognized as central players in a number of autoimmune diseases including SLE (see He J, *et al. Immunity.* 2013 Oct 17; Shane Crotty, *Immunity.* 2014 Oct 16). These abnormalities can be responsible for *Pten* deficient platelets-induced autoimmunity.

We have demonstrated that *Pten* deficient platelets secreted more inflammatory and Tfh promoting cytokines by activating the PDK1/mTORC2-AKT-SNAP23 signaling (Fig. 4 d-g). We agree with the reviewer that identification of other intrinsic mechanisms would be important for our understanding how *Pten* deficient platelets induced autoimmunity. As we hope the referee is aware, characterization of detailed mechanism, at least to date, is a publication by itself. It is, however, simply beyond our capabilities at this point within a reasonable timetable. While, we are working on this as we also think this is a really interesting and fruitful area of further research.

3. PTEN deficient platelets can cause autoimmune phenotypes are novel findings but It is not clear which immune cell compartment(s) are a primary contributor. Excessive Tfh activation could be a secondary effect. More comprehensive studies are needed to

be done as the reviewers suggested.

Response: We agree that it is unclear that which cell type is a primary responsible for *Pten* deficient platelet-caused autoimmunity in our study. The autoimmune phenotypes are heterogeneous, and the immune homeostasis is maintained by coordination of multiple types of innate and adaptive cells. To investigate which immune cell type is another primary contributor, we analyzed young *Pten*^{fl/fl}*Pf4-Cre* mice (about 6 to 8-weeks-old) which have not developed lymphoproliferative and autoimmune diseases. Most of mice displayed excessive Tfh cell responses, but have normal other CD4⁺ T helper subsets and CD8⁺ T cell. These observations have been mentioned in the revised manuscript (page 7, line 116-118). Conversely, some mice first developed more Th1 and Th17 cells, but still have normal Tfh cells. An increase of Th1, Th17 or Tfh cell alone can initiate the autoimmunity diseases, and secondly affects other immune cell types. Considering that the effects of platelets on the differentiation of other CD4⁺ T helper lineages have been established (see Norbert Gerdes, *et al. Thromb Haemost.* 2011; Eugene D. Ponomarev, *Front. Immunol.* 2018; Nailin Li, *Thromb Haemost.* 2013; L Zhu, *et al. J Thromb Haemost.* 2014), we focus on how platelet educated Tfh cells in this study. As we stated in the manuscript that *Pten*-deficient platelets cause lymphoproliferative and autoimmune diseases through programming multiple types of adaptive and innate immune cells (page 15, line 293-301). Thus, it is impossible to determine the most primary contributor to autoimmunity in the content of *Pten* deficient platelets.

Tfh responses can occur prior to the onset of autoimmunity (page 7, line 116-118), and *Pten*-deficient platelets enhanced their ability of infiltration into the lymphoid tissues, interaction with CD4⁺ T cells and promoting Tfh cell differentiation under co-culture or platelet transfer conditions (revised Fig. 5 and 6), providing the evidence that *Pten*-deficient platelets can directly drive Tfh cell differentiation. However, *Pten*-deficient platelets can regulate the maturation and cytokine secretion of dendritic cells (see Elzey BD, *et al. Immunity.* 2003), and may subsequently induce excess Tfh cells. We also have stated in the revised manuscript that *Pten*-deficient platelets mediate Tfh cell differentiation and further autoimmunity via direct and indirect

mechanisms (page 14, line 281-285).

4. The disease relevance of this study is weak. More data from suitable patients or adding more data from autoimmune diseases models would be informative.

Response: This is a good suggestion. It is interesting to elevate the association of inflammatory cytokines with hyperactive platelets or PTEN malfunction in SLE patients. But we have not attempted to address this in a substantive manner. In part, this is because of the difficulty in obtaining enough clinical samples or suitable autoimmune diseases models for analysis during the revision of this manuscript. As suggested by the editor, we have discussed the clinical relevance of our study in the discussion section of the revised manuscript (page13, line 244-258).

REVIEWER COMMENTS

Reviewer #3 (Remarks to the Author):

The authors have presented the arguments for the concerns raised by the reviewer.

I agree with the authors' recognition of the limitation of their study and modified the title of this manuscript accordingly.

But lack of evidence for determining the most primary contributor to autoimmunity in the content of Pten deficient platelets is a major defect of this article.

1. It is also very confusing that the authors mentioned most mice displayed excessive Tfh cell responses, but have normal other CD4+ T helper subsets and CD8+ T cells. Conversely, some mice first developed more Th1 and Th17 cells, but still have normal Tfh cells.

2. Another critical issue is the lack of pathophysiological significance of Pten deficient platelets mediated autoimmunity in humans. Authors had discussed that mutations within the PTEN gene have been reported in SLE patients by whole-exome sequencing but if these mutations are loss of function or gain of function are not clear and most importantly these germline mutations could affect all immune cells but deficiency of PTEN on different cell types can cause opposite effects. It is difficult to explain biological impacts of this gene on the development of autoimmune diseases.

3. On discussion part authors mentioned that "notably, Pten-deficient platelets may regulate other immune cells such as dendritic cells, which are critical for the initiation of Tfh cell differentiation, to indirectly induce excessive Tfh cell responses" , Do authors have some unpublished data or other papers support this point?

It is necessary to measure the DC phenotype in the early stage of platelet-specific pten-deficient mice.

Point-by-point responses to the reviewer's comments:

We would like to thank the editor and reviewer for their positive and encouraging comments as well as their constructive criticisms. The changes we have made are highlighted by red in the revised manuscript.

To Reviewer #3:

The authors have presented the arguments for the concerns raised by the reviewer.

I agree with the authors' recognition of the limitation of their study and modified the title of this manuscript accordingly. But lack of evidence for determining the most primary contributor to autoimmunity in the content of *Pten* deficient platelets is a major defect of this article.

Response: We thank the reviewer for his /her appreciation of improvement of revised manuscript. Our new experiments provide new evidences to show that Tfh cells are the most primary contributor to autoimmunity observed in platelet-specific *Pten*-deficient mice (see below).

1. It is also very confusing that the authors mentioned most mice displayed excessive Tfh cell responses, but have normal other CD4⁺ T helper subsets and CD8⁺ T cells. Conversely, some mice first developed more Th1 and Th17 cells, but still have normal Tfh cells.

Response: Differentiation of each CD4⁺ helper subset is controlled by a specific set of cytokines. *Pten*-deficient platelets secreted a variant of inflammatory cytokines, including IL-6, IL-21 and IFN- γ . Differential combinations of these cytokines induce the different differentiation fate of CD4⁺ helper cells. Accumulating evidences have demonstrated that platelets are heterogeneous populations, and contain different secretory granules subtypes (see Heijnen, *et al. Journal of Thrombosis and Haemostasis*, 2015; Wang, *et al. Cell Stem Cell* 2021). In addition, platelet secretion is kinetically heterogeneous in an agonist-response manner (see Jonnalagadda, *et al. Blood*, 2012). It is possible that platelet-specific *Pten*-deficient platelets create different profiles of inflammatory cytokines across distinct tissues and mice. Tfh-prone cytokine environment is prevailing in the context of *Pten* deficient platelets,

probably accounting for the observation that most mice displayed excessive Tfh cell responses, but still have normal other CD4⁺ T helper subsets and CD8⁺ T cells.

2. Another critical issue is the lack of pathophysiological significance of Pten deficient platelets mediated autoimmunity in humans. Authors had discussed that mutations within the PTEN gene have been reported in SLE patients by whole-exome sequencing but if these mutations are loss of function or gain of function are not clear and most importantly these germline mutations could affect all immune cells but deficiency of PTEN on different cell types can cause opposite effects. It is difficult to explain biological impacts of this gene on the development of autoimmune diseases.

Response: There were two reported SLE patients with germ-line mutation of PTEN, and one case harbored truncating mutation (c.697C > T, p.R233X), which is expected to be loss of function (see Al-Mayouf, *et al. Clin Rheumatol*, 2020; Tirosh I, *et al. Pediatr Rheumatol Online J*, 2019). Germ-line *Pten*^{+/-} mice developed a polyclonal autoimmune disorder (see Di Cristofano, *et al. Science*, 1999). Later studies revealed that *Pten* deficiency in T cells contributes to the autoimmune disease. Our current study further demonstrates that *Pten* expression in platelets is also required for restraining autoimmunity. These findings imply that germ-line mutations of PTEN in human cause autoimmunity in part by regulating the phenotypes of T cells and platelets, although the effects can be partially counteract by PTEN deficiency in other immune cells. Germ-line PTEN mutations are rare in SLE patients based on literatures, and we think that PTEN malfunction in platelets are most likely caused by transcriptional and/or post-translational mechanisms. We are collecting samples to test this notion.

3. On discussion part authors mentioned that "notably, Pten-deficient platelets may regulate other immune cells such as dendritic cells, which are critical for the initiation of Tfh cell differentiation, to indirectly induce excessive Tfh cell responses" , Do authors have some unpublished data or other papers support this point?

It is necessary to measure the DC phenotype in the early stage of platelet-specific pten-deficient mice.

Response: We thank the reviewer for this important comment. As suggested by the

reviewer, we analyzed the DC phenotype in 3- to 5- month-old *Pten^{fl/fl}Pf4-Cre* mice. Although most *Pten^{fl/fl}Pf4-Cre* mice have developed excess Tfh cells in the spleens, they displayed normal dendritic homeostasis (revised Supplementary Fig. 4, Page 7, line 118-123). These results provide the evidence to show that *Pten*-deficient platelets induce excess Tfh cells prior to disruption of DCs. Accordingly, we have rectified previous notion in the revised manuscript (Page 14-15, line 287-292). In combination with previous observation that Tfh responses can occur prior to the onset of autoimmunity (Page 7, line 116-118), we propose that excess Tfh cells are the most primary contributors to autoimmunity in *Pten^{fl/fl}Pf4-Cre* mice (Page 14-15, line 287-292).